

# Influence of solar variability on the occurrence of Central European weather types from 1763 to 2009

Mikhaël Schwander[1,2], Marco Rohrer[1,2], Stefan Brönnimann[1,2], Abdul Malik[1,2]

[1]Institute of Geography, University of Bern, Bern, 3012, Switzerland
[2]Oeschger Centre for Climate Change Research, University of Bern, Bern, 3012, Switzerland

*Correspondence to*: Mikhaël Schwander (mikhael.schwander@giub.unibe.ch)

**Abstract.** The impact of solar variability on weather and climate in Central Europe is still not well understood. In this paper we use a new time series of daily weather types to analyse the influence of the 11-year solar cycle on the tropospheric weather of Central Europe. We employ a novel, daily weather type classification over the period 1763-2009 and investigate
the occurrence frequency of weather types under low, moderate and high solar activity level. Results show a tendency towards fewer days with westerly and west south-westerly flow over Central Europe under low solar activity. In parallel, the occurrence of northerly and easterly types increases. Changes are consistent across different sub-periods. For the 1958-2009 period, a more detailed view can be gained from reanalysis data. Mean sea level pressure composites under low solar activity also show a reduced zonal flow, with an increase of the mean blocking frequency between Iceland and Scandinavia. Weather
types and reanalysis data show that the 11-year solar cycle influences the late winter atmospheric circulation over Central Europe with colder (warmer) conditions under low (high) solar activity. Model simulations used for a comparison do not reproduce the imprint of the 11-year solar cycle found in the reanalyses data.

## 1. Introduction

The effects of solar activity changes and on weather and climate in Europe are still not well understood. Although there is
both empirical and model evidence of an imprint of the 11-yr sunspot cycle in the stratosphere, climate effects at the Earth's surface are less clear, nor are the mechanisms understood. Considering the rather small changes in the incoming energy over an 11-yr sunspot cycle of ca. 0.1% (and perhaps also over longer periods), many of the suggested mechanisms are indirect and involve changes in atmospheric circulation (see Gray et al., 2010, for a review). Therefore, analysing changes in atmospheric circulation with regard to the 11-yr sunspot cycle (and perhaps longer-term changes) might help to better
attribute climatic changes to solar forcing. In this paper we analyse the imprint in atmospheric circulation.

Solar activity can have effects on the atmospheric circulation through three different mechanisms. These effects may arise from direct changes in total solar irradiance (TSI), from changes in stratospheric ozone induced by changes in solar UV, or from changes in stratospheric ozone induced by energetic particles, whose flux is modulated by solar activity. The ~1 Wm$^{-2}$ variation in TSI over an 11-yr sunspot cycle corresponds to a change in the radiation forcing of about ~0.17 Wm$^{-2}$ (Haigh



2003; Gray et al., 2010). This change in radiation forcing is estimated to cause a change in Earth's surface temperature of approximately 0.07 K and - with a lagged response - to changes in sea-surface temperatures (SSTs) (Gray et al., 2010; Stevens and North, 1996; White et al., 1997). Circulation effects may arise from unequal heating or ocean feedbacks that might involve the North Atlantic (Thiéblemont et al., 2015). The increased UV radiation during sunspot maxima leads

directly to an ozone increase and associated heating in the upper stratosphere (~40 km) (e.g., Matthes et al., 2004; Sitnov, 2009) and to changes in tropospheric circulation via downward propagation from the stratosphere. The suspected effects project strongly onto the North Atlantic European sector (Baldwin et al. 2001). The energetic particle flux (proton and electron), which is peaking in the declining phase of the sunspot cycle, leads to the production of $NO_x$ in the mesosphere and stratosphere, which can destroy ozone in the stratosphere (Andersson et al., 2014; Päivärinta et al., 2013; Solomon et al.,

1982). Through downward propagation, the troposphere can be affected, but a phase lag is expected. The mechanisms might lead to different temporal (i.e., lagged or not) or spatial circulation changes, hence it is important to well characterise the circulation response of the 11-year solar cycle.

For all of the mechanisms, the response is expected to be pronounced over the North Atlantic European sector. In fact, many observation-based studies have found effects of the 11-year sunspot cycle in European weather and climate (e.g.,

Barriopedro et al., 2008; Brugnara et al., 2013; Huth et al., 2007; Ineson et al., 2011; Lockwood et al., 2010). The impact of solar activity on variability modes such as the Atlantic Oscillation (AO) or the North Atlantic oscillation (NAO) is often investigated. The AO - which is correlated with the NAO - was shown to be influenced in his intensity and variability by the 11-year solar cycle (Huth et al., 2007). The NAO was found to be linked to the 11-year cycle with a positive (negative) pattern being associated to high (low) solar activity (e.g., Gimeno et al., 2003; Ineson et al., 2011; Lockwood, 2012; Sfîca et

al., 2015; Woollings et al., 2010). Ineson et al. (2011) found a respond to low solar activity in models which is similar to the negative NAO pattern. A similar pattern under low solar activity was found by Woollings et al. (2010). Brugnara et al. (2013) did not find a significant correlation between the solar activity and the NAO, although they found a reduced westerly flow over the North Atlantic under low solar activity. Thiéblemont et al. (2015) found the NAO/solar activity coupling to be the strongest with a 3-year lag. A similar lag was found in Gray et al. (2013). The correlation between solar activity and the

NAO is not supported by all studies. van Oldenborgh et al. (2013) found no statistically significant linear relation between the sunspot number and the NAO.

The North Atlantic circulation shows a response to the 11-year cycle, which leads to changes in the European weather that could only be visible on short time-scales. Atmospheric circulation over Europe is strongly correlated to the NAO and hence solar activity is thought to have an influence on weather conditions in Europe in winter. Studies show a preference of cold

winters in Europe to be associated with minima in the 11-year solar cycle (e.g., Lockwood et al., 2010; Sirocko et al., 2012). Changes in the atmospheric circulation over the North Atlantic linked to solar activity might have an impact on European weather on short time-scale. For example Barriopedro et al. (2008) analysed the duration in days of blockings linked to solar



activity. They found that North Atlantic blocking persistence increases under low solar activity and they are positioned more to the east.

Model simulations have also been used to investigate the solar activity impact on climate (see Gray et al., 2005 for a review). Uncertainties are however still large concerning the response in the troposphere. Models can reproduce the main influence of

the solar activity on the troposphere but have difficulties to reproduce details. For instance, Gray et al. (2013) found a lag in the solar response over Europe and the North Atlantic in observation data which was not confirmed by model simulations. Gray et al. (2013) also concluded that there is no consensus between climate models on the influence of the 11-year solar cycle and the linked mechanisms. Matthes et al. (2003) compared the response of several global circulation models to the 11-year solar signal over Europe and the North Atlantic. One of their conclusions was that the late winter dynamical response in

the model is not comparable to observations.

Others studies looked at the impact of solar activity on climate at longer time-scales. Martin-Puertas et al. (2012) used lake sediments to analyse variations in wind strength and the $^{10}$Be accumulation rate for solar activity from 3300 to 2000 years before present. They found windy conditions in Western Europe during late winter under a long period of low solar activity. Moffa-Sánchez et al. (2014) used foraminifer shells to reconstruct the sea surface temperature and salinity of the North

Atlantic over the past millennium. They found a correlation between centennial-scale variations in hydrography and total solar irradiance. On a shorter time-scale, Sirocko et al. (2012) analysed the occurrence of cold winters in Europe back to 1780 using documentary data. Sirocko et al. (2012) found cold winters in Europe to be often linked to the low activity phase of the 11-year solar cycle. The time resolution of these studies covering a long period is coarse (centennial-scale) or in the case of Sirocko et al. (2012) the method shows some weaknesses as explained in van Oldenborgh et al. (2013).

In this study, we analyse the influence of the 11-year solar cycle on Central European weather types. The aim is to identify how the variations in the mean atmospheric circulation over Europe can be explained by changes in the occurrence of weather types. For this we apply a similar approach as Huth et al. (2008b) by looking at the occurrence of weather types over Central Europe. There is a large panel of weather type classifications (WTCs) available for Europe based on various methods and covering different periods (Huth et al., 2008a; Philipp et al., 2010, 2014). Here we use a unique data set of daily weather

types covering the period 1763-2009 (Schwander et al., accepted). It allows us for the first time to investigate the impact of the 11-year solar cycle on European climate with an analysis of weather statistics. This analysis is performed by looking at changes in weather type occurrence as well as within-type changes. We complete this analysis by comparing changes in reanalyses data with model simulations.

This paper is structured as follow. The data and the methods used to analyse the solar activity influence on weather types,

reanalysis data and model simulations are explained in Section 2. The results are presented in Section 3 and discussed in Section 4. We conclude this work in Section 5.



## 2. Data and Methods

For our analysis of the impact of 11-year solar cycle, we first computed the mean differences between low and high solar activity for the sea level pressure (slp), 500 hPa geopotential height (z500), 850 hPa temperature (t850) and blocking frequency for the period 1958-2009. For this, we used the ERA-40 (Uppala et al., 2005) and ERA-Interim (Dee et al., 2011) reanalyses data set (same method as in Section 2.5) and the monthly sunspot number as a measure of solar activity (Section 2.1). A two-tailed Student's t test was used to determine the 99% probabilities that the low and high solar activity composites are from two different populations. Then we extended the analysis by using weather types (Section 2.2) with focus and the occurrence changes (Section 2.3) and within-type differences (Section 2.4). The method for the blocking frequency used in the differences composites is described in Section 2.5. Finally, the mean differences computed from reanalyses were compared with model simulations (Section 2.6).

### 2.1 Solar activity

The monthly sunspot number is used as a measure of the 11-year solar cycle. It is the longest record of solar variability available; daily sunspots data are available starting from 1818 but monthly or yearly data go back to 1700. Sunspots correspond to zones of strong magnetic field; therefore, many visible sunspots are synonym of an active sun as the quiet sun is free of any spot. The sunspots time series captures well the 11-year solar cycle and can therefore be used as a proxy for quantifying solar activity. However, one limitation is that the sunspot number cannot be negative. The sun's activity might still vary even if no sunspots are visible; all sunspots minimum are not necessarily of the same TSI level (see Fig. 4). The sunspots data were retrieved from the Sunspot Index and Long-term Solar Observations (SILSO) and World Data Center for the production, preservation and dissemination of the international Sunspot Number website from the Royal Observatory of Belgium. Additionally to the Sunspot Number, we also used a TSI reconstruction from Shapiro et al. (2011). This reconstruction was used as forcing in the four model simulations used in this study (see Section 2.6).

### 2.2 Weather type classification

Weather types are used to determine if the circulation differences observed in reanalyses are only due to changes in the mean circulation or if they result from a change in the occurrences of weather patterns. For this, we use the CAP7 (cluster analysis of principal components) WTC (Schwander et al., accepted). CAP7 is a daily time series of weather types representing the mean atmospheric circulation in the Alpine Region and Central Europe over the period 1763-2009. This classification is a reconstruction of the CAP9 classification used by the Federal Office of Meteorology and Climatology MeteoSwiss (Weusthoff, 2011). CAP9 starts in 1957 and is updated to the present. The weather types were computed with the ERA-40 and ERA-Interim reanalyses dataset. CAP9 was used as referenced from 1958 to 1998 and was reconstructed back to 1763 using early instrumental data from European weather stations. The classification was reduced to 7 types (hence CAP7) by combining similar, not well discriminated types. Although CAP9 was originally computed for the Alpine Region and



contains a limited number of patterns, it – as well as CAP7 – captures the main circulation patterns over Europe and the North Atlantic. The 7 types with their abbreviations are presented in Table 1. The mean frequency of occurrence of each type for January to March (JFM) over the period 1763 to 2009 is shown in Fig. 1. The z500 and slp composites for 1958-2009 computed with ERA-40/-Interim for JFM are shown in Fig. 2. CAP7 is the only objective times series of daily weather types

which covers almost 250 years in Europe. It also covers a longer period than any existing reanalysis (from which weather types can also be computed). For more information on the method of reconstruction, see Schwander et al. (accepted). The daily weather types CAP7 are completed with a probability value of each day being correctly classified (relative to the reference classification). This allows us later to omit days with a probability lower than a certain threshold (e.g. 75%).

## 2.3 Weather type occurrences

The following method is similar to the procedure applied in Huth et al. (2008b) but the dataset used here covers a longer period of time. A comparison with Huth et al. (2008b) can however be done over the second part of the 20[th] century. To capture the influence of the sun on weather patterns, changes in the frequency of occurrence of the CAP7 weather types relative to variations in solar activity are analysed. It was shown that the strongest influence of solar activity on the low troposphere is visible during the late winter months because of the delayed propagation of the signal from the stratosphere to

the troposphere (Ineson et al., 2011). Thus, the weather type analysis as well as reanalyses and model simulations analyses are performed on months of JFM. The sunspot number data was first divided in three categories, low, moderate and high solar activity using the 33[rd] and 66[th] percentile as thresholds. This method assumes that all solar minima reach a similar low intensity as the number of sunspot cannot go below zero. The daily weather types were then classified to the corresponding solar activity level. For each weather type we computed the ratios of the frequency for each solar activity level (low,

moderate, high) relative to the long-term mean. Results are calculated for the period January 1763 to March 2009 as well as for five sub-periods of approximately 50 years in length. We removed 3 years following large volcanic eruptions as they can have a significant influence on climate (e.g., Robock, 2000). The list of volcanic eruption was taken from Arfeuille et al. (2014). A resampling method was used to test the significance of the ratios The weather type series (for each period) was resampled 10000 times. The computed ratio is considered as significant when below (above) the 250[th] (9750[th]) value of the

resample elements. Another series of histogram was computed using only weather types having a probability (to be correctly classified) superior to 75%. We also computed the ratios with a 1, 2 and 3 years lag for the 1763-2009 period.

## 2.4 Within-type differences

In addition to the change in the weather types' occurrences, we investigate the within-type difference of atmospheric fields between low and high solar activity for each of the 7 weather types. For this, we computed composites for each weather type

in order to identify changes in their mean circulation pattern over Europe and the North Atlantic under low and high solar activity. Composites of slp, z500 and t850 of each type were computed for the previously defined high and low solar activity classes. Additionally to these parameters, the mean blocking frequency was also computed for each composite (see Section



2.5). From these composites, differences were calculated by subtracting the high activity from the low activity composites. ERA-40 (1958-August 2002) and ERA-Interim (September 2002-2009) were used as the original CAP types were computed based on the ERA mean slp field. The within-types analysis can therefore only be made over the period 1958-2009 and cannot be extrapolated back to 1763.

**2.5 Blocking Frequency**

Blockings are defined as reversal of the meridional geopotential height gradient at 500 hPa. We follow the approach of Tibaldi and Molteni (1990) and extended the blocking algorithm to find blockings in a two dimensional field following the procedure of Scherrer et al. (2006). The algorithm flags a certain longitude and latitude as blocked, if two criteria are fulfilled:

1. GPH gradient towards the pole

$$GPHG_P = Z500(\varphi + 14) - \frac{Z500(\varphi)}{\varphi + 14 - \varphi} < -10\frac{gpm}{°lat} \tag{1}$$

2. GPH gradient towards the equator

$$GPHG_E = Z500(\varphi) - \frac{Z500(\varphi - 14)}{\varphi - \varphi - 14} > 0\frac{gpm}{°lat} \tag{2}$$

The latitude $\varphi$ varies from 36° to 76° in 2° intervals. ERA-40 data is bilinearly interpolated to a 2°x2° before computation. Only blockings with a minimum lifetime of 5 days and spatial overlap larger than 70% between each time step are considered here.

**2.6 Model simulations**

In the last part of this paper, we used model simulations to complete the previous analysis on the mean differences between low and high solar activity. The SOCOL-MPIOM model (Muthers et al., 2014) was used for this analysis. The SOCOL (Solar Climate Ozone Links) chemistry-climate model is coupled to the ocean-sea-ice model MPIOM. SOCOL is based on the middle atmosphere model MA-ECHAM5 version 5.4.01 (Roeckner et al., 2003) and a modified version of the chemistry model MEZON (Model for Evaluation of oZONe trends, Egorova et al., 2003). Several major external forcings were applied
in the transient simulations. It includes radiative forcing from major greenhouse gases (CO2, CH4, N2O and CFCs). The volcanic forcing is computed as global annual mean aerosol optical depth in the visible band. The TSI was calculated from the SSI reconstruction of Shapiro et al. (2011). In addition, the upper envelope of the uncertainty range was taken as moderate solar activity (smaller amplitude forcing) in the simulation. This reconstruction differs from previous ones (Schmidt et al., 2012) because of its larger amplitude. It has the advantage to be a predominant forcing in the model with
visible impacts in the simulations. The Sunspot Number is one of the proxies used in this reconstruction. In addition to the impact of the high frequency (11-year cycle), we can use the model also to look at effects of low frequency solar activity





(prolonged periods of low and high solar activity) by comparing simulations in which only the low-frequency component of the solar forcing changed. The results can then be compared back to those obtained from reanalysis data. We want to see of the model reproduce similar changes in the tropospheric weather in Europe (slp and t850) linked to the 11-year cycle. Also we can compare the impact of the 11-year cycle (Fig. 3) to the impact of the low frequency of the solar activity (i.e. grand

minimum, Fig. 4). For more information on the SOCOL-MPIOM model see Muthers et al. (2014).

We used four model simulations covering the period 1600-1999, two with a large solar activity amplitude (L1 and L2, different initial conditions, Shapiro reconstruction) and two with a moderate amplitude (M1 and M2, different initial conditions) which correspond to the upper bound of the Shapiro reconstruction uncertainty. For the analysis, we removed again 3 years following large volcanic eruptions (Arfeuille et al., 2014). Note that many of the important eruptions occur

during a solar minimum. The 11-year solar cycle was analysed similarly as for the weather types (33[rd] and 66[th] thresholds). Two periods were selected, 1958-1999 - for comparison with the reanalysis data - and 1763-1999. The low frequency solar activity (Shapiro, Fig. 4) was again divided again using the 33[rd] and 66[th] percentile threshold (with respect to the period 1600-1999). High and low solar activity composites were computed for slp and t850, and the low minus high activity difference was calculated. TSI forcing is correlated with the anthropogenic forcing ($CO_2$, $CH_4$, $N_2O$ and CFCs) with an

increase over the 19[th] and 20[th] century. Both forcings reach their highest values at the end of the 20[th] century (see Muthers et al., 2014). We removed the anthropogenic forcing by applying a linear regression:

$$y_i = \alpha + \beta x_i + \varepsilon \qquad (3)$$

where $y$ is the predicted value and $x$ the predictor, $\alpha$ the intercept, $\beta$ the regression coefficient and $\varepsilon$ the residual.

## 3. Results

### 3.1 Mean difference

The differences computed between low and high solar activity with ERA-40 and ERA-Interim for 1958-2009 (Fig. 5) show a reduced zonal flow over Europe under low solar activity relative to high activity. The slp is higher between Iceland and Scandinavia, and lower over Southern Europe and the Mediterranean Sea. The z500 differences have a similar pattern but with higher values which extend more to the west over Greenland. The blocking frequency is also higher over this region

under low solar activity especially between Iceland, the northern British Isles and western Scandinavia. The higher values extend also to the south-western part of Europe. The t850 is reduced over most of the European continent and North Africa, and higher between Greenland, the northern British Isles and Scandinavia.

### 3.2 Solar signal in the occurrence of the weather types

The frequencies of occurrence of CAP7 weather types for different solar activity levels for JFM are shown in Fig. 6. The

histograms display the ratios computed between the low, moderate and high solar activity frequencies and the long-term



mean frequency. Histograms (a) to (e) correspond each approximately to a 50-year period (1763-1807, 1808-1857, 1858-1907, 1908-1957, 1958-2009) and histogram (f) to the whole time series (1763-2009). Histogram (e) (1958-2009) correspond to the reanalysis period and roughly to the period (1950-2002) analysed in (Huth et al., 2008b). For these last 50 years, the Northerly (N), North-Easterly (NE) and Westerly flow over Southern Europe (WC) types have the highest ratios under low solar activity but only the Northerly type ratio is significantly different from 1. At the same time, the West South-Westerly (WSW) and High Pressure (HP) types have the lowest ratios (non-significant). Under high solar activity, the Northerly (N) type have a ratio significantly lower than 1. Under medium activity, no ratio is significantly different than 1.

For the other periods of time (sub-periods (a) to (e)) the ratios have a large variability in-between them. For example, in none of them the Northerly (N) type ratio is significantly higher than 1. This kind of variability is visible in most of the weather types. Another example is the North-Easterly (NE) type under low solar activity where the ratio is lower (higher) than 1 in sub-periods (b) and (c) ((a) and (d)). There are only two weather type which has stable ratios over time. Under low solar activity the Westerly (W) and West South-Westerly (WSW) types ratios are lower than 1 in four of the five sub-periods.

Although it can be difficult to deduce a general structure (similar ratios under the same solar activity level) in the weather types occurrences between the different sub-periods (a) to (e), there are some significant changes in the mean occurrence of some of the types over the whole time series (1763-2009, (f)). Under low solar activity, we observe significantly lower ratios of West South-Westerly (WSW) and Westerly (W) types, and significant higher ratios of High Pressure (HP) type. Under moderate solar activity the higher West South-Westerly (WSW) and Westerly (W) types ratios are significant, as well as the lower High Pressure (HP) type ratio. Finally, under high solar activity the higher Westerly (W) type ratio and is significant.

Ratios in Fig. 7 were computed only with days with a probability higher than 75%, allowing us to omit some potentially erroneous weather type data. Again there is a large variability in-between the different sub-periods. Over the period 1763-2009, minor changes appear compared to Fig. 6. The higher ratio in the Easterly type (E) under low solar activity is significant. Under moderate solar the higher West South-Westerly (WSW) type ratio is not significant. Hence the ratio for the same type (WSW) is higher than 1 (but not significant) under high solar activity.

The sub-period (a) (1763-1807) shows some of the largest differences between the three solar activity classes (Figs. 6 and 7). The sub-period 1763-1807 is shorter than all other ones and therefore contains fewer days (especially for Fig. 6). It is also the period in which the weather types' reliability is the lowest.

The decrease in the occurrence of the Westerly type (W) is also visible with a 1, 2 and 3 years lag (not shown), as for the West South-Westerly (WSW) it is only visible with a 1 year lag. The increase in the occurrence of these two types under high solar activity is visible with a 1 and 2 years lag but disappear with a 3 years lag. It is even significant for the 2 years lag. The signal found in the Northerly type (N) is inverted with a 2 and 3 years lag with a reduction (increase) in the occurrence under low (high) solar activity. The increase in the occurrence of the High Pressure type (HP) under low solar activity is the strongest with a 3 year lag.



The main occurrence differences can be summarised as follows: The occurrence of Westerly and West-South Westerly types decreases under low solar activity relative to high activity. At the same time, we observe a higher occurrence of High Pressure, Northerly and Easterly types. The occurrences under moderate activity are similar to those observed under high activity. The number of days with a Westerly and West-South Westerly (High Pressure and Easterly) type increases (decreases).

### 3.3 Solar signal in weather types – within-type differences

The inter-type analysis is completed with a within-type analysis of their composites (Fig. 8 & 9). Difference composites were computed by subtracting the high from the low solar activity class composites. They were computed for the period 1958-2009 with ERA-40/Interim and are therefore not representative of the whole 1763-2009 period. Fig. 8 displays the z500 and blocking frequency differences. Fig. 9 displays the slp and t850 differences. The weather types were originally computed with the slp over the Alpine region; thus the smallest slp differences are expected to be observed over this region. However, differences appear in the position and intensity of the high and low pressure centres, this can influence the general flow and thus the temperatures over Europe and the Alpine region.

With Fig. 8 and 9 we can identify the influence of the solar activity on each weather type and from this try to deduce a general influence on the tropospheric weather over Europe. The following descriptions always refer to the low solar activity class composite relative to the high activity one. The mean weather types slp and z500 composites are shown in Fig. 2.

1 (NE): The low pressure system south of Greenland extends more to the south and less from Iceland to Scandinavia. The anticyclone is weaker over Western Europe but extends more towards Scandinavia. The low pressure system over Italy is slightly deeper. The blocking frequency is higher from Greenland to Scandinavia but lower further south over the Atlantic. These changes in the pressure pattern lead to lower temperature over the whole European continent.

2 (WSW): The low pressure system located between Iceland and Scotland is less pronounced over Northern Europe. The mean z500 is higher between the British Isles and Scandinavia, and lower over Eastern Europe and the Mediterranean. The same pattern is visible in the temperature differences. Small differences in the blockings with higher frequencies over Scandinavia are also visible

3 (W): The pressure over Iceland is reduced whereas the Azores anticyclone is more pronounced over the Atlantic, the pressure gradient is tighter. The pressure is higher over Scandinavia and the anticyclone is more present over Southern Europe with a higher blocking frequency. Temperatures are therefore lower over most parts of Europe.

4 (E): The pressure is higher between Greenland and Scandinavia as well as the blocking frequency. The anticyclone extends more over Europe and the pressure is lower over the Mediterranean Sea. The temperature is reduced over all Europe except Scandinavia.





5 (HP): The pressure and z500 are higher over most of the North Atlantic and Northern Europe, whereas lower over Southern Europe and Northern Africa. The blocking frequency is higher over all Europe. The temperature is reduced over Europe especially in the eastern part.

6 (N): The pressure and z500 are higher over Scandinavia and lower over the Western Mediterranean Sea and Eastern Atlantic. The flow is more oriented north-easterly than north-westerly over Central Europe with reduced temperature.

7 (WC): The Azores anticyclone is more pronounced and the low pressure system between the British Isles and Scandinavia is weaker but extend more towards the Mediterranean Sea. The temperatures are reduced over South-Eastern Europe and Northern Africa, whereas warmer over North-Eastern Europe.

Similar patterns can be observed among the weather types. Types 1 (NE), 4 (E) and 6 (N) all have an enhanced easterly flow over central Europe under low solar activity and thus lower temperatures, All three types also have more frequent Scandinavian Blockings. Types 2 (WSW) and 3 (W) have a slightly reduced westerly flow over Central Europe. On average (ALL on Figs. 8 & 9, also Fig. 7) we see a higher pressure between Iceland and Scandinavia, and lower pressure over the Mediterranean Sea under low solar activity. The blocking frequency is higher between Iceland and Scandinavia too. This leads to a weaker pressure gradient and westerly flow over Europe. Following this reduction in the zonal flow, temperatures tend to be lower over Europe (except Scandinavia). Outside Europe we note an increase in temperature over the high latitude in all cases especially around Greenland.

### 3.4 Solar signal in model simulations

The model simulations are used to complete the analysis of the weather types and the reanalysis data. The differences between low and high solar activity obtained by four simulations are displayed in Fig. 10, 11 and 12. In Fig. 10 and 11 the low and high solar activity classes corresponds to the 11-year solar cycle (Fig. 3). In Fig. 12 the classes correspond to extended periods of weak and strong activity (Fig. 4). M1 and M2 are the simulations with a moderate solar activity amplitude whereas L1 and L2 correspond to the simulations with a large amplitude (Fig. 3). Again the high solar activity is subtracted from the low activity and the slp and t850 differences are shown.

The difference plots in Fig. 10 should be comparable to Fig. 5 as well as the "ALL" plot in Fig. 9. However, none of the four simulations display a similar difference pattern as the reanalysis data. There is a lower pressure over the North Atlantic (M1), Scandinavia (L1), and Eastern Atlantic/Western Europe (L2) under low solar activity, whereas M2 shows a higher pressure and temperature over Europe. Only M1 have extended lower temperature over Europe, but the slp differences do not fit with the reanalyses data.

Over the period 1763-2000 (Fig. 11) the differences between low and high solar activity are similar but less pronounced as in Fig. 10. One exception is the L1 simulation which has a higher pressure over Scandinavia under low solar activity. Lower



temperatures over Eastern Europe can also be seen. This pattern resembles that the mean slp difference in the reanalyses (Fig. 5 and "ALL" in Fig. 9).

In Fig. 12 (low frequency solar influence), all four simulations have a similar pattern of differences. M1 shows higher pressure values between Iceland and Scandinavia. The pressure is reduced over the Atlantic between 25° N and 50° N. M2 is

similar with a pattern shifted to the north-west with positive differences extending more towards Greenland and negative differences covering part of Europe. L1 is similar to M2 over the Atlantic and Western Europe with larger values. Finally L2 is very similar to M1 with positive values between Greenland and Europe and slightly negative values over the Atlantic. M1 and L2 have a similar slp pattern as found in the reanalyses data (Fig. 5 and ALL in Fig. 9). 850 hPa temperatures are cooler under low solar activity over all Europe except around the Iberian Peninsula. We see also that this reduced westerly flow has

consequent cooler temperature over Europe. The reduction in temperature is probably a combination between changes in circulation and reduced solar radiations.

## 4. Discussion

The reduced zonal flow and colder temperatures over Europe under low solar activity are consistent with other studies (e.g., Brugnara et al., 2013; Ineson et al., 2011; Sfîca et al., 2015; Sirocko et al., 2012; Woollings et al., 2010). The differences

over Europe resembles a more negative (positive) NAO pattern under low (high) solar activity. This kind of pattern was suggested in several studies (e.g., Ineson et al., 2011; Sfîca et al., 2015; Thiéblemont et al., 2015). However, our results over the North Atlantic correspond more to a positive NAO under low solar activity relative to high activity with a lower slp south of Greenland and a stronger Azores high pressure system. So it seems that the 11-year cycle does not directly modulate the NAO but shows more a west-east pattern between the Labrador Sea and Western Russia. Other studies corroborate this

pattern with a solar signal extending toward Eurasia (Brugnara et al., 2013; Woollings et al., 2010). For Brugnara et al. (2013) the Eurasian index is more linked to the 11-year cycle than the NAO. The differences in the blockings index confirm the reduced zonal flow under low solar activity with a higher blocking frequency over the Norwegian Sea and Scandinavia. Similarly, Barriopedro et al. (2008) found an increase in the blocking frequency over the Eastern Atlantic under low solar activity. They also found an increase in the blocking frequency under high solar activity over the Western Atlantic which we

did not find in our results.

The differences in weather type occurrences over the period 1958-2009 correspond well with the results of Huth et al. (2008b). It is especially the case for the West South-Westerly type with a decrease in their occurrence under low solar activity. The Northerly type shows also the same pattern in Fig. 4 and 5 as in Huth et al. (2008b) with an increase (decrease) in the occurrence under low (high) solar activity. The differences in these two types (WSW and N) are also confirmed in the

long-term (1763-2009) differences in our results. Under low (high) solar activity the frequency of occurrence of Westerly and West-South Westerly types decreases (increases). This consecutively results in an increase in the frequency of Easterly





and High Pressure types under low solar activity. The reduction in the occurrence of Westerly and West South-Westerly is the largest difference in the ratios between one solar activity class and the long-term mean that we observe and is visible over almost all the shorter periods of time. There is however a large variability in the variations in the occurrences. Certain types (e.g. North-Easterly) show a large variability in their occurrence across time and no mean pattern can be identified over

the whole period analysed. These types could be more sensitive to internal variability or to the influence of others forcings. As a counterexample it is interesting to see that in almost all cases we observe a decrease in the occurrence of the Westerly type under low solar activity. This persistence of this signal over time supports the hypothesis that the 11-year solar cycle has an influence on the occurrence of European weather types.

A reduction (increase) in the occurrence of westerly types under low (high) solar activity as well as an increase in the

occurrence of easterly types under low activity leads to a decrease (increase) in temperature. In addition, these changes in the pattern of weather types occurrences are consistent with a weaker (stronger) zonal flow over the North Atlantic and Europe, and a negative (positive) NAO phase pattern under low (high) solar activity as it was suggested in several studies (e.g., Ineson et al., 2011; Sfîca et al., 2015).

A lagged response of the NAO following a solar maximum was suggested in Gray et al. (2013) and Thiéblemont et al.

(2015). Our results with a 1 and 2 years lag show a higher occurrence of westerly types under high solar activity also support the hypothesis of a delayed signal. However, we do not observe any signal with 3 years lag under high solar activity. Under low solar activity the weather types occurrences are similar at with no lag and a 1 year (reduction in westerly and west south-westerly types, increase in easterly, northerly and high pressure types.) Although for the westerly and high pressure types the signal is still visible with a 2 and 3 years lag, it gets inverted for the westerly, northerly and high pressure types

The mesoscale circulation variations can explain the changes in the frequency of occurrence of the Easterly and Westerly weather patterns over Central Europe. A weaker zonal flow leads to a reduction of the occurrences of westerly types and thus to an increase in the occurrence of easterly types (continental flow). These pattern changes are also consistent with a higher blocking index over Scandinavia under low solar activity. Blockings over high European latitudes are often responsible for the establishment of an easterly flow over Central Europe. We also observe – from 1958 to 2009 – not only a change in the

weather types occurrences but also on the slp patterns of each weather type. For the Westerly and West-South Westerly types we observe a reduction in the slp between Greenland and Iceland under low solar activity. Similarly to the mean difference (no weather type discrimination), it resembles more to a positive NAO phase which is in contradiction with previous studies (e.g., Ineson et al., 2011; Thiéblemont et al., 2015). However, further east (toward Scandinavia) the pressure is higher under low solar activity which is synonym of a reduced oceanic flow over Central Europe and lower temperatures. So it seems that

the 11-year cycle does not directly modulate the NAO but shows more a west-east pattern between the Labrador Sea and Western Russia. As mentioned above, other studies corroborate this pattern with a solar signal extending toward Eurasia (Brugnara et al., 2013; Woollings et al., 2010).



As mentioned above, an increase in slp and blockings over Scandinavia as well as a decrease in slp over the Mediterranean Sea are synonym of an enhanced continental flow over Central Europe. We notice a double effect with an increase in the occurrence of Easterly and Northerly types (inter-type) under low solar activity but also a stronger mean easterly flow based on their composites for the period 1958-2009 (within-type). The same holds for Westerly and West South-Westerly types,

which are less frequent and the associated zonal flow to these patterns is also slightly weaker over Central Europe. The stronger (weaker) continental (zonal) flow under low solar activity brings cold air from the Eurasian continent and diminishes the influence of the warm oceanic air over Central Europe. Following these circulations changes we estimate that there is a higher (lower) probability to have cold winter during the weak (strong) phase of the 11-year solar cycle. Other studies (Lockwood et al., 2010; Sirocko et al., 2012) found similar results with cold European winters being often linked to

weak solar activity.

The comparison with model simulations does not (or only partially) confirm our observation-based results on the mean slp over the North Atlantic and Europe. The response of the slp and t850 to the 11-year solar cycle does not display any clear pattern, each simulation having a different response. The low amplitude of the 11-year cycle in the Shapiro TSI combined with the relatively coarse resolution of the model could explain the difficulty of the model to capture changes over a specific

region. Also, during phases of grand minima the 11-year cycle almost vanishes. Even if the 11-year solar cycle is visible after 1958, the simulations do not show any signal in slp similar to the reanalysis data. The differences between grand minima and maxima (low frequency) are closer the reanalysis data (high frequency, 11-year cycle differences). The model captures well the general cooling linked to the reduced solar forcing but also displays slp differences which are similar to ERA-40/-Interim with higher slp over the North Atlantic/Scandinavia and therefore a weaker zonal flow under low solar

activity.

## 5. Conclusion

We have used a new weather types classification to analyse the impact of the 11-year solar cycle on European weather in late winter. The monthly sunspot number was used as a measure of solar activity and the daily weather types were retrieved from the CAP7 classification. We have analysed the frequency of occurrence of the weather types under three different solar

activity levels (low, moderate, high) from 1763 to 2009 and analysed as well as the within-type differences between low and high solar activity from 1958 to 2009 in reanalyses data. The mean difference in the sea level pressure and 850 hPa temperature was then compared with model simulations.

The strongest solar signal visible in the occurrence of the CAP7 weather types is a reduction in the number of days with westerly and west south-westerly flow under low solar activity. Consequently, we observe an increase in days with a

northerly, easterly flow and high pressure. Conversely, the occurrence of both westerly and west south-westerly types increases under moderate and high solar activity. Not only the occurrence of some weather types respond to change in the

solar activity, but also the mean pattern of these types are slightly different. We observe on average a weaker zonal flow over Europe under low solar activity for westerly types and a stronger continental flow for easterly, north-easterly and northerly types. This is also confirmed by the higher blocking frequency over Scandinavia under low solar activity. The sea level pressure differences observed in the reanalysis data are not supported by model simulations. But we estimate that the

SOCOL-MPIOM model is not ideal for an analysis of the 11-year solar cycle impact on tropospheric weather. However, we suggest that the differences between prolonged period of low and high solar activity are similar to the 11-year response.

The 247-year long analysis of the 11-year solar cycle impact on late winter European weather patterns suggest a reduction in the occurrence of westerly flow types linked to a reduced mean zonal flow under low solar activity. Following these observation, we estimate the probability to have cold conditions in winter over Europe to be higher under low solar activity

than under high activity. Also similar conditions can occur during periods of prolonged reduced total solar irradiance.

**Acknowledgments.** This work was funded by the Swiss National Science Foundation through the Sinergia FUPSOL II (Number 147659). We thank ECMWF for providing ERA-40 and ERA-Interim data. We also thank the World Data Center for the production, preservation and dissemination of the international Sunspot Number.

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



**Table 1: CAP7 weather types numbers, abbreviations and names**

| Index | Abbreviation | Full Name |
|:---:|:---:|:---|
| 1. | NE | North-East, indifferent |
| 2. | WSW | West South-West, cyclonic, flat pressure |
| 3. | W | Westerly flow over Northern Europe |
| 4. | E | East, indifferent |
| 5. | HP | High Pressure over Europe |
| 6. | N | North, cyclonic |
| 7. | WC | Westerly flow over Southern Europe, cyclonic |

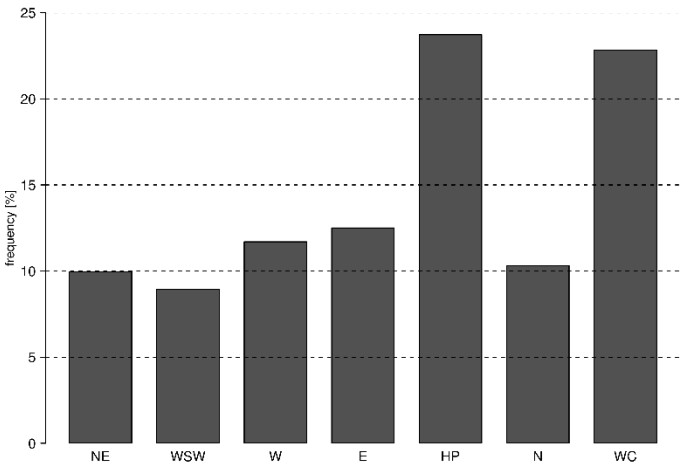

5   **Figure 1: CAP7 1763-2009 JFM mean frequency of occurrence.**

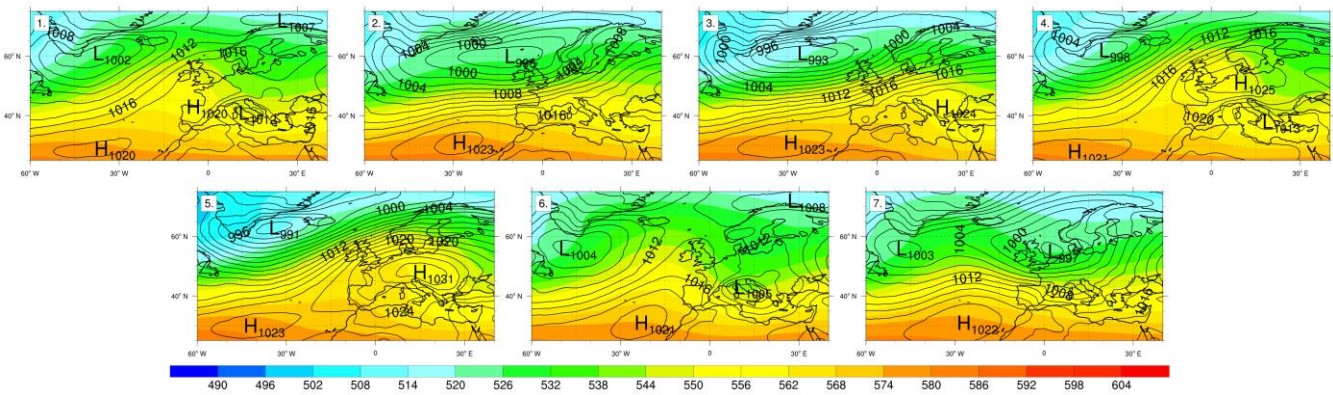

**Figure 2: CAP7 1958-2009 500 hPa geopotential height (color) and sea level pressure (contours) JFM composites.**



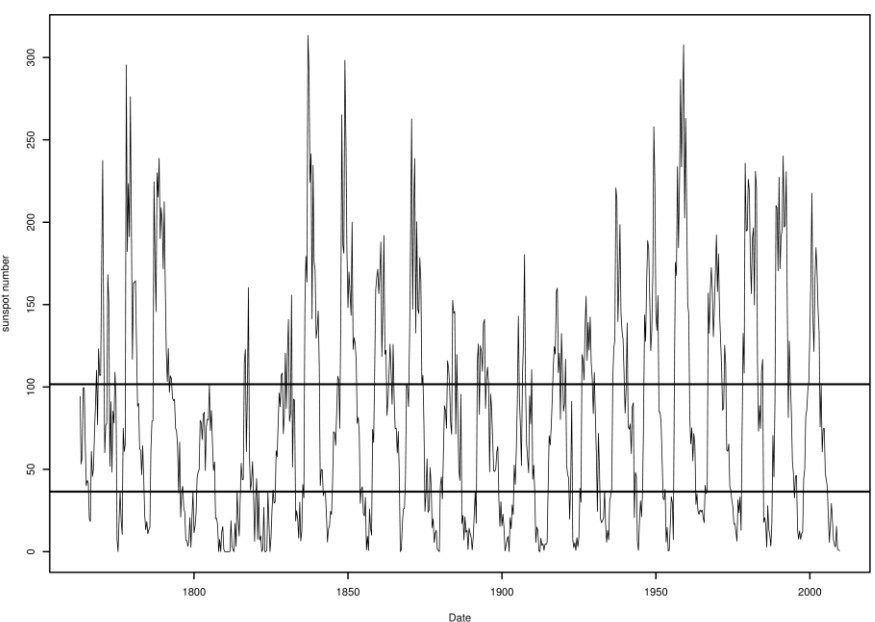

**Figure 3: 1763-2009 JFM monthly Sunspot Number with 33$^{rd}$ and 66$^{th}$ precentile thresholds.**

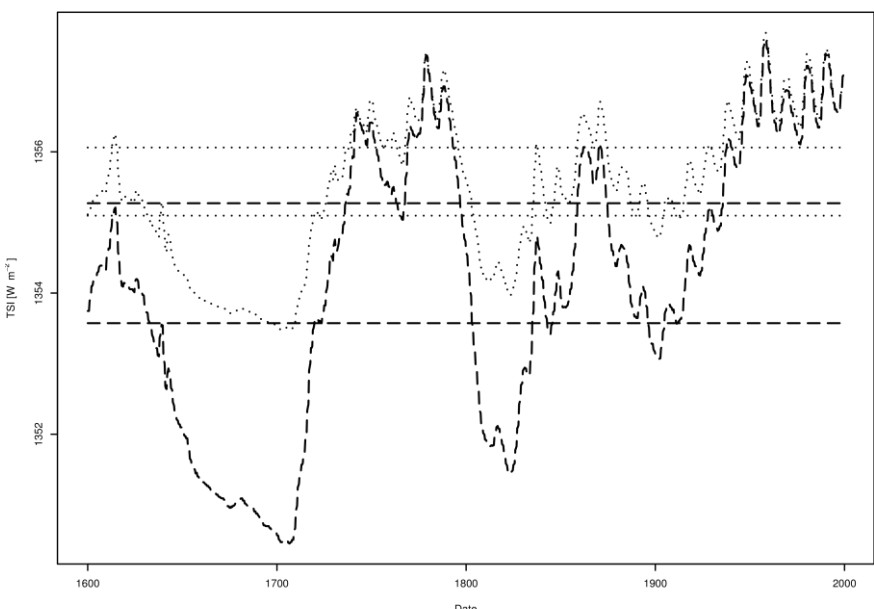

5    **Figure 4: 1600-1999 TSI (W m$^{-2}$) retrieved from Shapiro solar activity reconstruction. Dotted line correspond to the medium amplitude (simulations M1/M2) and dashed line correspond the large amplitude (simulations L1/L2), with corresponding 33$^{rd}$ and 66$^{th}$ percentile thresholds.**



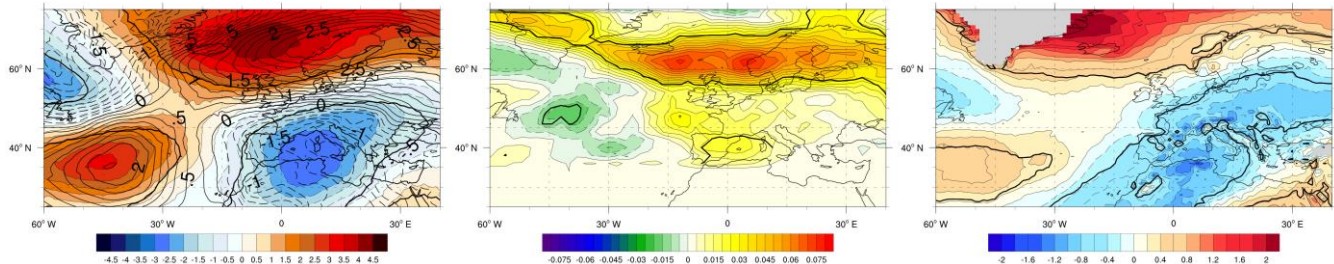

**Figure 5: 1958-2009 low minus high solar activity differences computed with ERA-40/-Interim. Left: 500 hPa geopotential height (color) and sea level pressure (contour). Centre: blocking frequency. Right 850 hPa temperature. The 99% significance level is indicated with bold black line.**



**Figure 6: Ratios of the frequency for the low (light grey), moderate (grey) and high (black) solar activity classes for different periods. Dots correspond to statistical significance of the ratios at the 95% level.**



**Figure 7: Ratios of the frequency (probability of weather types >75%) for the low (light grey), moderate (grey) and high (black) solar activity classes for different periods. Dots correspond to statistical significance of the ratios at the 95% level.**





**Figure 8: CAP7 (1 to 7) and mean (ALL) blocking frequency (color) and 500 hPa geopotential height (contour) difference between low and high solar activity (low minus high) computed with ERA-40/-Interim for 1958-2009.**





**Figure 9: CAP7 (1 to 7) and mean (ALL) sea level pressure (contour) and 850 hPa temperature (colour) difference between low and high solar activity (low minus high) computed with ERA-40/-Interim for 1958-2009.**





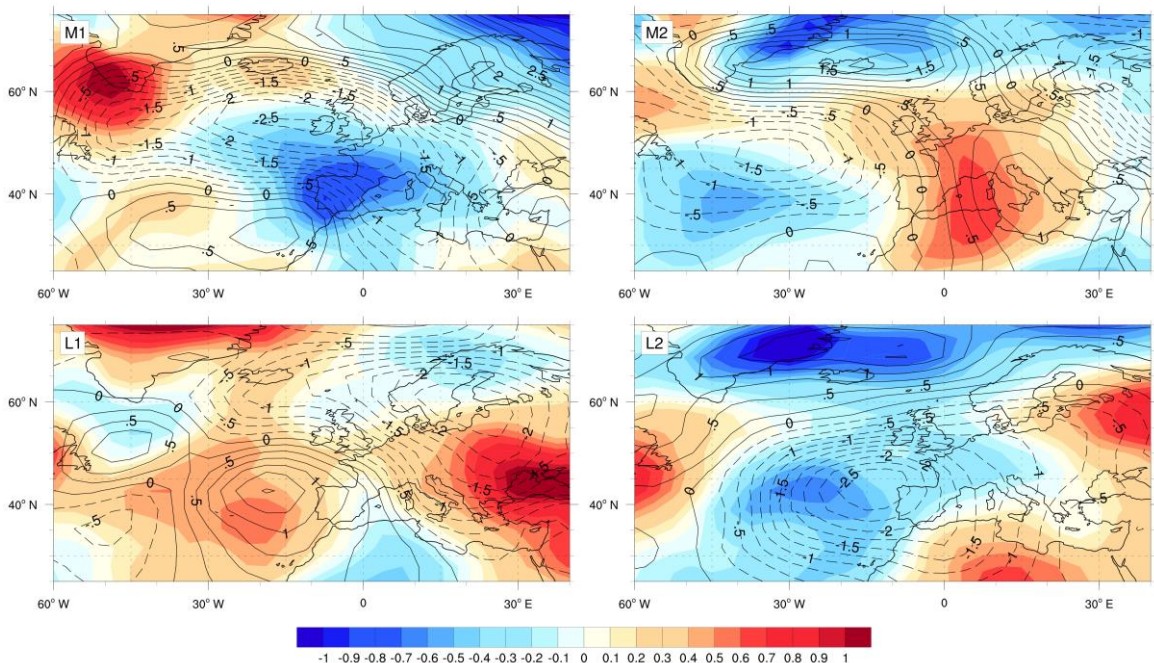

**Figure 10: Sea level pressure (contour) and 850 hPa temperature (colour) difference between 11-year cycle low and high solar activity (high frequency) computed with the model simulations for 1958-2000.**

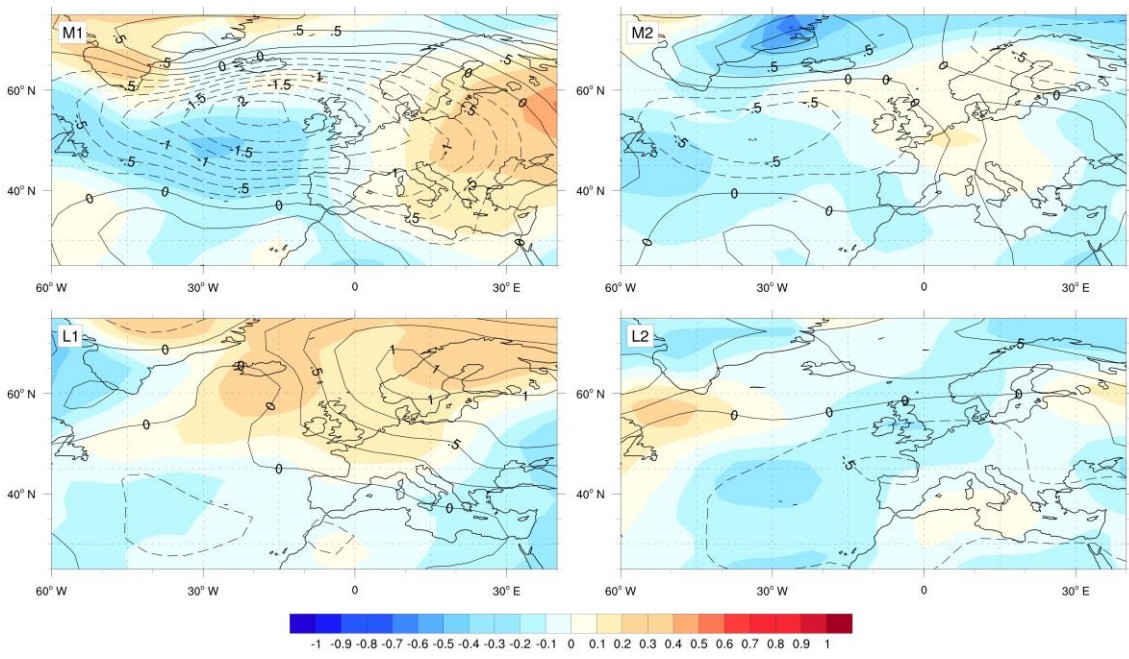

**Figure 11: Sea level pressure (contour) and 850 hPa temperature (colour) difference between 11-year cycle low and high solar activity (high frequency) computed with the model simulations for 1763-2000.**





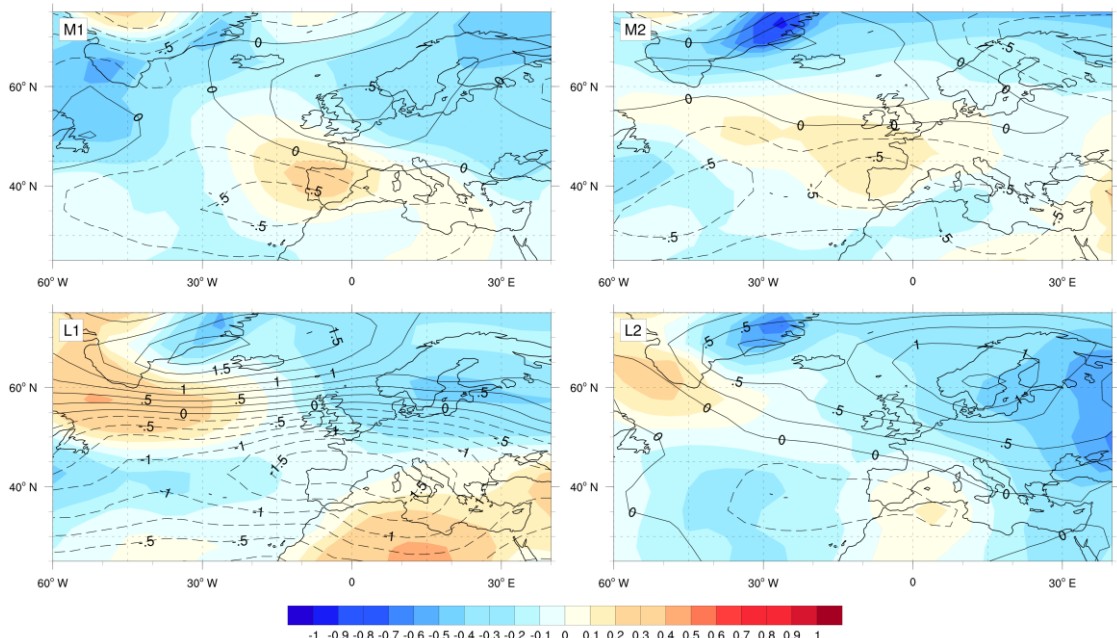

**Figure 12: Sea level pressure (contour) and 850 hPa temperature (color) difference between low and high solar activity (low frequency) computed with moderate (M1/M2) and large (L1/L2) solar activity simulations for 1600-1999.**

