# Peer review of "Influence of solar variability on the occurrence of Central European weather types from 1763 to 2009"

_Climate of the Past, 2017_

## Referee Comment (RC1) · Anonymous Referee #1 · 28 Feb 2017

The manuscript presents a detailed investigation of 11-yr solar cycle influence on weather types. The topic is of interest for the paleoclimate and solar-terrestrial communities, although a detailed knowledge of the weather types and their variability may be too farfetched for most of the CP readers. Some description of weather type characteristics is needed. Besides a new, unique reconstruction of weather types, the authors analyze a set of 4 simulations with a climate model forced by Total Solar Irradiance, only. This is a somewhat a strange model configuration and weakens the merit of the simulations for detecting mechanisms, as the "top-down", which believed to be the key mechanism influencing the weather types (see Introduction) is neglected. Most parts of the manuscript are well written and reading is straightforward and easy, except the

discussion part where it is difficult to pair results of this study to the text. There are some points other that need further clarification and improvement before publication of the paper.

Methodology

Weather types are analyzed by the means of composites. The 11-yr solar cycle is sliced in three groups/day of low, moderate and high activity. Are these groups of near equal size? I am concerned about the role of internal variability in the composites and how affects results. How confident are the authors that compositing results in true solar signals? Splitting the record to 50 yr chunks offers too little to this regard because it provides little evidence of consistency over time. This is recognized by the authors: "P8 L13 Although it can be difficult . . .". I would suggest to split to two sub-periods at best. For the same reason, Figure 7 can be supplied as a supplementary.

Significance in solar minimum

I always consider the solar minimum as the least perturbed state of climate not necessarily the reverse of the solar maximum. It is puzzling to me that the most noticeable changes in WSW and W types are detected in solar minimum, when the forcing is weakest. Could the authors elaborate on the reasons/mechanisms that can explain strongest signals in solar minimum and not maximum?

Model simulations

Perhaps I am missing something here, but my understanding is that the SOCOL simulations are forced only by TSI and in particular by the strong Sapiro et al. TSI reconstruction. There is nothing wrong by choosing a strong TSI reduction to facilitate the signal-to-noise detection. My objection here is on the specification of TSI and not SSI variability. Is there any particular reason to assume that solar signals in weather types are attributed to the "bottom-up" mechanisms? Most of the discussion in introduction emphasizes the importance of "top-down" mechanisms in transferring signals on the

surface, a mechanism which apparently is missing in model runs without SSI forcing. In such a case, the low resemblance between reanalysis and modelled signals is not surprising to my understanding. Moreover, some similarities discussed in P.10 L30 is a matter of coincidence to me. So, it is difficult to understand the overall point of Section 3.4 given that the SOCOL runs are missing key mechanisms. The weakness of the simulations should be discussed in the text.

Mean difference (Section 3.1)

Do results of figure 5 compare with Fig1 of Ineson et al., 2011? Difficult to say for SLP. For temperature, I see some similarities but some differences as well. I could also consider presenting lagged anomalies (see my following comment)

Weather type classification (Section 2.2)

This section assumes a reader familiar with the different weather types and their within-type differences. I am afraid this won't be the case for most of the CP readers. For example, what does the "well discriminated types (P4 L 31)" mean? Or, "days with probability higher than 75%". I think a concise description of the main characteristics of the weather types is needed.

Lagged responses

The authors in P12 3rd paragraph, briefly discuss the lagged response of westerly types and try to compare with Gray et al. and Thieblemont et al., results. Same in P8 last paragraph. Inferring time lags is very interesting subject and I would recommend a proper presentation, dedicating, perhaps, even a new Section. This could be a valuable contribution to the number of recent papers discussing time lags as they can highlight the importance of atmosphere-ocean coupling.

Some additional considerations,

P1. L27: stratospheric ozone + "and heating"

P2. L10: "phase lag is expected": Perhaps this is not true by the sole action of "top-down" mechanisms. An atmosphere-ocean coupling is required for lags longer than one year at least.

P2 L20: found a response

P3. L3: do you mean Gray et al., 2010?

P3 L4: This is hardly true. Gray et al, show surface signals.

P3. I think the second paragraph should also be extended by discussing results of more recent model intercomparison such as CCMVal or SolarMIP. See (Austin et al., 2008; Hood et al., 2015; Misios et al., 2015; Mitchell et al., 2015) and references therein.

P3 L25: "It allows us . . . weather statistics". Is this true? What is the main difference to Huth et al., 2008b?

P4 L4: Description here is rather confusing. You should clarify that you analyze a merged dataset and not ERA-40 and ERA-int separately. Please elaborate how stitching was performed.

P4 L17 : Is it one of the revised products of sunspot numbers?

P4 L29: "from 1958 to 1998". Why not till 2009?

P6 L25: $CO_2$, $CH_4$, $N_2O$ (subscripts)

P7 L6: A quantitative difference of the forcing, long term and 11-yr cycle, should be given here.

P7 L10: . . .66th thresholds of sunspot numbers?

P7 L11: Still not clear how percentiles are calculated. Have you subtracted the 11-yr solar cycle before?

P11-13: It is very difficult to follow the discussion of the results. Please point to the associated figures.

P13, L11: "only partially". This is a wishful thinking!

Figure 5: Difficult to separate SLP from geopotential signals. Please consider splitting this panel in two.

Austin, J. et al., 2008. Coupled chemistry climate model simulations of the solar cycle in ozone and temperature. J Geophys Res-Atmos, 113(D11): D11306. Hood, L. et al., 2015. Solar Signals in CMIP-5 Simulations: The Ozone Response. Q. J. Roy. Meteorol. Soc., 141: 2670–2689. Misios, S. et al., 2015. Solar Signals in CMIP-5 Simulations: Effects of Atmospherre Ocean Coupling. Q. J. Roy. Meteorol. Soc. Mitchell, D.M. et al., 2015. Solar Signals in CMIP-5 Simulations: The Stratospheric Pathway. Q. J. Roy. Meteorol. Soc., 141: 2390-2403.

---

## Referee Comment (RC2) · Anonymous Referee #2 · 10 Mar 2017

Review of the paper "Influence of solar variability on the occurrence of Central European weather types from 1763 to 2009" by Mikhaël Schwander et al., MS No.: cp-2017-8

General comments

The paper uses a novel weather type classification that was constructed by the authors in a previous recent study, in order to identify and assess the potentially important regional aspects of solar variability effects on the weather types in central Europe for the period from 1763 to 2009. The present paper expands the use of the weather type classification and contains new material.

However, the paper needs major improvements before it is considered for publication.

The authors try to assess and compare the shorter term (11years) solar variability effect to the long-term (secular and super secular) changes, occurring at periods of 90-years or more. This attempt is not very successful, as it is not clear throughout the paper where they discuss which time scale. All sections of the paper, mainly the introduction, the data sections and the discussion on the model study, and, of course the conclusions, should be rewritten so that the paper's message is conveyed clearly to the reader. Suggestions on major issues are given below.

Specific Comments

The introduction section is rather poor on bibliography, and could be enriched more;. e.g.on page 3, line 5 they refer to Gray et al., 2005, an older paper compared to Gray et al., 2010. Moreover, they should at least mention the work of Meehl et al., 2009, or van Loon and Meehl, Seppala et al., 2009, Rozanov et al., 2012, Scaife et al., 2013, and at least refer to the work by e.g. Mitchell et al., 2015, Misios et al., 2016 on the solar signal in the CMIP5 simulations. (A relatively recent review on the mechanisms and effects is given also by Seppälä et al, 2014)

The data section is incomplete. In the very first paragraph they mention that they used ERA-40 and ERA-Interim. My impression is that these two reanalysis data sets have been used as one. However, it is not clear if this is the case and, if yes, if there has been any check done on the homogeneity of the data, or if the possible discrepancies have been identified and corrected.

Section 2.1 should be clearly written, and the indices they used for the 11-year and longer term variability presented in a very clear way. For example, there is no call to Figure 3 in this section. They refer to Figure 4 but with no explanation as to what it contains, and the reader is left puzzled, since the Shapiro reconstruction is shown there without it being mentioned in the text. Moreover, I could not understand why they mention in the text that the fact that the sunspot cycle does not become negative is a

limitation (this is also mentioned again later in the paper).

Section 2.3 It is not clear what are the time scales they discuss. Do they refer to the 11-year of the secular cycles? This should be very clearly mentioned here as well. The mechanism they refer to is the top-down mechanism, in which the stratospheric response and the signal transfer from there to the troposphere is the main pathway. This leads us to

Section 2.6, where they describe the model simulations. Again in line 21 they refer to low and high solar activity, with no clear indication as to what they mean. Moreover, and for the model simulations: Was TSI the only forcing? Or did they use also the appropriate SSI forcing? Was the model run in its full version with the interactive ozone response in the stratosphere? How is it achieved if one uses TSI variations only? Was the solar effect on ozone included in any way? If SSI variability with the solar cycle and the stratospheric response is not included, then one can have only the bottom-up mechanism, and the comparison to e.g. Ineson et al. is not straight forward. In addition, what is the meaning of " It has the advantage to be a predominant forcing in the model.."? It is also not clear how the 11-year solar cycle is handled here. The Shapiro index and its use to define "large solar activity", "moderate amplitude" should be more clearly written.

Page 7 line 9-10, on the volcanic activity and the years that were removed. Why do you state there to "note that many of the important eruptions occur during a solar minimum". Is there any possible connection? How does the removal affect your statistics if it was mainly done for solar minimum years? And more importantly, what type of solar minimum? Sunspot, or secular?

Page 7, lines 15 -18. How exactly was the anthropogenic forcing removed? What were the predictors? Was there only one predictor? Which one?

Section 3.3 Significance in the differences should be given. The same holds for every place where differences are discussed.

4 Discussion Page 11, lines 18-19. It is accepted that the 11-year cycle effects project onto tropospheric circulation patterns like the Arctic Oscillation (AO) and the North Atlantic Oscillation (NAO) rather than are directly correlated to NAO or AO

5. Conclusions page 14, lines 4-6. The present simulation and the forcings used (if indeed SSI variability and ozone related variability have not been used) do not allow the investigation of the top-down mechanism, which is in the heart of the weather type response..

---

## Author Comment (AC1) · 19 Apr 2017

Response to:

Review of the paper "Influence of solar variability on the occurrence of Central European weather types from 1763 to 2009" by Mikhaël Schwander et al., MS No.: cp2017-8

General comments

The paper uses a novel weather type classification that was constructed by the authors in a previous recent study, in order to identify and assess the potentially important regional aspects of solar variability effects on the weather types in central Europe for

the period from 1763 to 2009. The present paper expands the use of the weather type classification and contains new material.

However, the paper needs major improvements before it is considered for publication. The authors try to assess and compare the shorter term (11years) solar variability effect to the long-term (secular and super secular) changes, occurring at periods of 90-years or more. This attempt is not very successful, as it is not clear throughout the paper where they discuss which time scale. All sections of the paper, mainly the introduction, the data sections and the discussion on the model study, and, of course the conclusions, should be rewritten so that the paper's message is conveyed clearly to the reader. Suggestions on major issues are given below.

Answer: We thank the reviewer for the constructive comments. Most parts of the papers will be improved and rewritten. We agree that some important information is missing to have a good understanding of the methods. The model simulations need to be better explained to explain since it also includes SSI. We will probably just focus in the 11-year solar cycle to be more consistent and leave out the low frequency solar variability since it does not add any relevant conclusions to the paper. The introduction and discussion on bottom-up and top-down mechanisms will also be improved.

Specific Comments

The introduction section is rather poor on bibliography, and could be enriched more;. e.g.on page 3, line 5 they refer to Gray et al., 2005, an older paper compared to Gray et al., 2010. Moreover, they should at least mention the work of Meehl et al., 2009, or van Loon and Meehl, Seppala et al., 2009, Rozanov et al., 2012, Scaife et al., 2013, and at least refer to the work by e.g. Mitchell et al., 2015, Misios et al., 2016 on the solar signal in the CMIP5 simulations. (A relatively recent review on the mechanisms and effects is given also by Seppälä et al, 2014).

Answer: We agree that some important references are missing and that some more recent papers should be cited. The introduction will be improved and completed with

more recent papers.

The data section is incomplete. In the very first paragraph they mention that they used ERA-40 and ERA-Interim. My impression is that these two reanalysis data sets have been used as one. However, it is not clear if this is the case and, if yes, if there has been any check done on the homogeneity of the data, or if the possible discrepancies have been identified and corrected.

Answer: We agree that some description of the data is missing. ERA-40 (from January 1958 to August 2002) and ERA-Interim (from September 2002 to March 2009) have been used together. The reason why ERA-40 was used until August 2002 and ERA-Interim from September 2002 is because the reference classification used to produce CAP7 (Schwander et al., 2017) was originally computed with ERA-40 (1958-2002) and ERA-Interim (2002-2009). For more information on the reference classification (CAP9) please see Weusthoff (2011). We tried to stay consistent with CAP7 (and therefore with CAP9) and use these two reanalysis dataset over the same two periods. The data were remapped to $1° \times 1°$ in order to be combined together. We have not found any discrepancies but we will compare both reanalysis over a similar period as an assessment.

Section 2.1 should be clearly written, and the indices they used for the 11-year and longer term variability presented in a very clear way. For example, there is no call to Figure 3 in this section. They refer to Figure 4 but with no explanation as to what it contains, and the reader is left puzzled, since the Shapiro reconstruction is shown there without it being mentioned in the text. Moreover, I could not understand why they mention in the text that the fact that the sunspot cycle does not become negative is a limitation (this is also mentioned again later in the paper).

Answer: We agree on the comment; this section will be improved and the corresponding figures will be mentioned and better presented.

Section 2.3 It is not clear what are the time scales they discuss. Do they refer to the

11-year of the secular cycles? This should be very clearly mentioned here as well. The mechanism they refer to is the top-down mechanism, in which the stratospheric response and the signal transfer from there to the troposphere is the main pathway. This leads us to

Answer: In Section 2.3 we speak about the 11-year sunspot cycle. It will be mentioned more clearly.

Section 2.6, where they describe the model simulations. Again in line 21 they refer to low and high solar activity, with no clear indication as to what they mean. Moreover, and for the model simulations: Was TSI the only forcing? Or did they use also the appropriate SSI forcing? Was the model run in its full version with the interactive ozone response in the stratosphere? How is it achieved if one uses TSI variations only? Was the solar effect on ozone included in any way? If SSI variability with the solar cycle and the stratospheric response is not included, then one can have only the bottom up mechanism, and the comparison to e.g. Ineson et al. is not straight forward. In addition, what is the meaning of "It has the advantage to be a predominant forcing in the model.."? It is also not clear how the 11-year solar cycle is handled here. The Shapiro index and its use to define "large solar activity", "moderate amplitude" should be more clearly written.

Answer: We realize that the description of the model simulations in the manuscript is not clear enough. SSI is included in the model. Here is a clearer description that we will include in the revised manuscript:

"In the present investigation we have employed the Coupled Atmosphere-Ocean-Chemistry Climate Model (AOCCM) simulations carried out with SOCOL-MPIOM (see, Muthers et al. 2014). The SOCOL (Solar Climate Ozone Links) chemistry-climate model is coupled to the ocean-sea-ice model MPIOM. The SOCOL is based on the middle atmosphere model MA-ECHAM5 version 5.4.01 (Roeckner et al., 2003) and a modified version of the chemistry model MEZON (Model for Evaluation of oZONe

trends, Egorova et al., 2003). The model has a horizontal resolution of T31 (3.75° × 3.75°) with 39 irregular vertical pressure levels (L39) from 1000 hPa to 0.01 hPa. The horizontal resolution of the ocean component (MPIOM) is 3o varying between Greenland (22 km) and tropical Pacific (350 km). The SOCOL-MPIOM cannot reproduce the Quasi-Biennial-Oscillation (QBO), thus nudged to QBO reconstruction from Brönnimann et al. (2007). The MA-ECHAM5 (MPIOM) component calculates the dynamical processes in every 15 (144) minutes and atmosphere-ocean coupling takes place in every 24 hours (Anet et al. 2013a, b; Muthers et al. 2014). Muthers et al. 2014 employed SOCOL-MPIOM to carry out four transient simulations (namely L1, L2, M1, and M2) over the period AD 1600-1999 with all major forcings (i.e. greenhouse gases, volcanic eruptions, aerosols, and solar spectral irradiance), and interactive ozone chemistry. The SOCOL-MPIOM was forced with six bands of Solar Spectral Irradiance (SSI) reconstruction of Shapiro et al. (2011) over the Ultraviolet (UV), visible, and near infrared ranges. The L1 (M1) and L2 (M2) simulations were forced with large (small) mean solar amplitude of 6 (3) W/m2 with different ocean initial conditions for both runs. For more details of the model the reader is referred to Muthers et al. 2014. The model is well capable of simulating the top-down (stratospheric-tropospheric coupling) and bottom-up (coupled ocean-atmosphere response) mechanisms as proposed by Meehl et al. (2009)."

"It has the advantage to be a predominant forcing in the model.." means that since the Shapiro reconstruction has a higher amplitude (∼6 w/m2) than any other reconstruction, it consists of a strong forcing in the model. The upper boundary of the uncertainty of the Shapiro reconstruction was used as moderate amplitude (∼3 w/m2) in the model. Also the Shapiro reconstruction includes the 11-yr solar cycle (based on the sunspot number) although it is often masked by the low frequency amplitude.

We agree that the use of model simulations should be better explained and justified in the paper. We will probably focus only on the period 1958-1999 in the model simulations as a comparison with the reanalysis data. Also the low frequency variability of

the solar variability during the period 1958-1999 is stable and we can focus only on the 11-yr solar cycle.

Page7 line9-10, on the volcanic activity and the years that were removed. Why do you state there to "note that many of the important eruptions occur during a solar minimum". Is there any possible connection? How does the removal affect your statistics if it was mainly done for solar minimum years? And more importantly, what type of solar minimum? Sunspot, or secular?

Answer: We are not aware about any connection between volcanic eruptions and the solar cycle. The fact that more volcanic eruptions occurred under low solar activity phases has mostly an impact on the size of (number of months) of the low solar activity class. The three solar activity classes (low, moderate, high) are not of the same size. There are 195 months under low solar activity, 211 under moderate activity, and 212 under high activity. We will add this information (number of months in each class) on the figures or in the text.

Page 7, lines 15 -18. How exactly was the anthropogenic forcing removed? What were the predictors? Was there only one predictor? Which one?

Answer: The predictor consists in the radiative forcing applied in the model calculated from major greenhouse gases ($CO_2$, $CH_4$, $N_2O$ and CFCs). They were taken from the PMIP3 database (Etheridge et al., 1996, 1998; Ferretti et al., 2005; MacFarling-Meure et al., 2006).

Section 3.3 Significance in the differences should be given. The same holds for every place where differences are discussed.

Answer: We agree, significance will be added on all figures and discussed in the text. Also it will be corrected in Figure 5 since we have found a small error in the significance plotted.

4 Discussion Page 11, lines 18-19. It is accepted that the 11-year cycle effects project

onto tropospheric circulation patterns like the Arctic Oscillation (AO) and the North Atlantic Oscillation (NAO) rather than are directly correlated to NAO or AO

Answer: We will reformulate the sentence to make it clear that we are not speaking about a direct correlation.

5. Conclusions page 14, lines 4-6. The present simulation and the forcings used (if indeed SSI variability and ozone related variability have not been used) do not allow the investigation of the top-down mechanism, which is in the heart of the weather type response..

Answer: SSI and the related ozone variability are included in the model (see above).

References

Anet. J. G., Muthers, S., Rozanov, E. V., Raible, C. C., Peter, T., Stenke, A., Shapiro, A. I., Beer, J., Steinhilber, F., Brönnimann, S., Arfeuille, F. X., Brugnara, Y., Schmutz, W.: Forcing of stratospheric chemistry and dynamics during the Dalton Minimum. Atmos Chem Phys 13:10951–10967. doi:10.5194/acp-13-10951-2013, 2013a.

Anet, J. G., Rozanov, E. V., Muthers, S., Peter, T., Brönnimann, S., Arfeuille, F. X., Beer, J., Shapiro, A. I., Raible, C. C., Steinhilber, F., Schmutz, W. K.: Impact of a potential 21st century "grand solar minimum" on surface temperatures and stratospheric ozone. Geophys Res Lett 40(16). doi:10.1002/grl.50806, 2013b.

Brönnimann, S., Annis, J. L., Vogler, C., Jones, P. D.: Reconstructing the quasi-biennial oscillation back to the early 1900s. Geophys Res Lett 34(L22805). doi:10.1029/2007GL031354, 2007.

Egorova, T., Rozanov, E., Zubov, V. and Karol, I. L.: Model for investigating ozone trends (MEZON), Izv. Atmos. Ocean. Phys., 39, 277–292, 2003.

Etheridge, D., Steele, L., Langenfelds, R., Francey, R., Barnola, J., and Morgan, V.: Natural and anthropogenic changes in atmospheric CO2 over the last 1000

years from air in Antarctic ice and firn, J. Geophys. Res., 101, 4115–4128, doi:10.1029/95JD03410, 1996.

Etheridge, D. M., Steele, L. P., Francey, R. J., and Langenfelds, R. L.: Atmospheric methane between 1000 A.D. and present: Evidence of anthropogenic emissions and climatic variability, J. Geophys. Res., 103, 15979–15993, doi:10.1029/98JD00923, 1998.

Ferretti, D. F., Miller, J. B., White, J. W. C., Etheridge, D. M., Lassey, K. R., Lowe, D. C., Meure, C. M. M. F., Dreier, M. F., Trudinger, C. M., Van Ommen, T. D., and Langenfelds, R. L.: Unexpected changes to the global methane budget over the past 2000 years, Science, 309, 1714–1717, doi:10.1126/science.1115193, 2005.

MacFarling-Meure, C., Etheridge, D., Trudinger, C., Steele, P., Langenfelds, R., Van Ommen, T., Smith, A., and Elkins, J.: Law Dome CO2, CH4 and N2O ice core records extended to 2000 years BP, Geophys. Res. Lett., 33, L14810, doi:10.1029/2006GL026152, 2006.

Meehl, G. A., Arblaster, J. M., Matthes, K., Sassi, F., Loon, H.V.: Amplifying the Pacific climate system response to a small 11-year solar cycle forcing. Science 325(5944). doi:10.1126/science.1172872, 2009.

Muthers, S., Anet, J. G., Stenke, A., Raible, C. C., Rozanov, E., Brönnimann, S., Peter, T., Arfeuille, F. X., Shapiro, A. I., Beer, J., Steinhilber, F., Brugnara, Y. and Schmutz, W.: The coupled atmosphere-chemistry-ocean model SOCOL-MPIOM, 5 Geosci. Model Dev., 7(5), 2157–2179, doi:10.5194/gmd-7-2157-2014, 2014

Roeckner, E., Bäuml, G., Bonaventura, L., Brokopf, R., Esch, M., Giorgetta, M., Hagemann, S., Kirchner, I., Kornblueh, L., 20 Rhodin, A., Schlese, U., Schulzweida, U. and Tompkins, A.: The atmospheric general circulation model ECHAM5: Part 1: Model description, MPI Rep., (349), 1–140, doi:10.1029/2010JD014036, 2003.

Schwander, M., Brönnimann, S., Delaygue. G., Rohrer. M., Auchmann, R., Brugnara,

Y. Reconstruction of Central European daily weather types back to 1763. International Journal of Climatology. doi: 10.1002/joc.4974. 2017.

Weusthoff, T.: Weather Type Classification at MeteoSwiss - Introduction of new automatic classification schemes, Arbeitsberichte der MeteoSchweiz, (235), 46, 2011.

---

## Author Comment (AC2) · 19 Apr 2017

Response to:

The manuscript presents a detailed investigation of 11-yr solar cycle influence on weather types. Thetopicisofinterestforthepaleoclimateandsolar-terrestrialcommunities, although a detailed knowledge of the weather types and their variability may be too farfetched for most of the CP readers. Some description of weather type characteristics is needed. Besides a new, unique reconstruction of weather types, the authors analyze a set of 4 simulations with a climate model forced by Total Solar Irradiance, only. This is a somewhat a strange model configuration and weakens the merit of the simulations for detecting mechanisms, as the "top-down", which believed to be the key

mechanism influencing the weather types (see Introduction) is neglected. Most parts of the manuscript are well written and reading is straightforward and easy, except the discussion part where it is difficult to pair results of this study to the text. There are some points other that need further clarification and improvement before publication of the paper.

Answer: We thank the reviewer for the constructive comments. We agree on most comments and will include the suggestions in the revised manuscript. The description of the model in our first manuscript is probably too vague and needs to be improved since the model is forced by SSI (see description below). We will probably leave out the low frequency solar variability and focus only on the 11-year solar cycle. The introduction and discussion on bottom-up and top-down mechanisms will also be improved as well as the description of the weather types.

Methodology Weather types are analyzed by the means of composites. The 11-yr solar cycle is sliced in three groups/day of low, moderate and high activity. Are these groups of near equal size? I am concerned about the role of internal variability in the composites and how affects results. How confident are the authors that compositing results in true solar signals? Splitting the record to 50 yr chunks offers too little to this regard because it provides little evidence of consistency over time. This is recognized by the authors: "P8 L13 Although it can be difficult...". I would suggest to split to two sub-periods at best. For the same reason, Figure 7 can be supplied as a supplementary.

Answer: The groups are not exactly of equal sizes since more volcanic eruptions occurred under low solar activity. There are 195 months under low solar activity, 211 under moderate activity, and 212 under high activity. We will add this information on the figures or in the text. The confidence is quite high since the signal in some types in significant over 250 years, and also because it is consistent with previous studies. The signal found in the occurrence of weather types is consistent with the within-type differences. The significance will be added on the within-type composites difference
plots.

We would keep the 1958-2009 sub-periods as a comparison with Huth et al. (2008). It is true that two sub-periods could make the results clearer and understandable. We will come to the same conclusions if we split in only two sub-periods. However, it would be more straightforward for the reader.

Significance in solar minimum I always consider the solar minimum as the least perturbed state of climate not necessarily the reverse of the solar maximum. It is puzzling to me that the most noticeable changes in WSW and W types are detected in solar minimum, when the forcing is weakest. Could the authors elaborate on the reasons/mechanisms that can explain strongest signals in solar minimum and not maximum?

Answer: We cannot provide any reasons why the signal could be stronger under low solar activity compared to high activity. We think that it comes from the fact that the long-term mean is also perturbed. If the low activity phase is not perturbed (one third of the months), then two thirds are perturbed (moderate and high). The long-term mean is therefore also perturbed and the differences under low solar activity seem larger. We could take the low solar activity phase as a (unperturbed) reference and then we would observe a strong increase in the occurrence of W and WSW types under moderate and high solar activity.

Model simulations Perhaps I am missing something here, but my understanding is that the SOCOL simulations are forced only by TSI and in particular by the strong Sapiro et al. TSI reconstruction. There is nothing wrong by choosing a strong TSI reduction to facilitate the signal-to-noise detection. My objection here is on the specification of TSI and not SSI variability. Is there any particular reason to assume that solar signals in weather types are attributed to the "bottom-up" mechanisms? Most of the discussion in introduction emphasizes the importance of "top-down" mechanisms in transferring signals on the surface, a mechanism which apparently is missing in model runs without

SSI forcing. In such a case, the low resemblance between reanalysis and modelled signals is not surprising to my understanding. Moreover, some similarities discussed in P.10 L30 is a matter of coincidence to me. So, it is difficult to understand the overall point of Section 3.4 given that the SOCOL runs are missing key mechanisms. The weakness of the simulations should be discussed in the text.

Answer: We realize that the model description in the Manuscript is lacking some information. The model was not forced by SSI and can include top-down and bottom-up mechanisms. The description of the model will be completed as follow in the revised manuscript:

"In the present investigation we have employed the Coupled Atmosphere-Ocean-Chemistry Climate Model (AOCCM) simulations carried out with SOCOL-MPIOM (see, Muthers et al. 2014). The SOCOL (Solar Climate Ozone Links) chemistry-climate model is coupled to the ocean-sea-ice model MPIOM. The SOCOL is based on the middle atmosphere model MA-ECHAM5 version 5.4.01 (Roeckner et al., 2003) and a modified version of the chemistry model MEZON (Model for Evaluation of oZONe trends, Egorova et al., 2003). The model has a horizontal resolution of T31 ($3.75° \times 3.75°$) with 39 irregular vertical pressure levels (L39) from 1000 hPa to 0.01 hPa. The horizontal resolution of the ocean component (MPIOM) is 3o varying between Greenland (22 km) and tropical Pacific (350 km). The SOCOL-MPIOM cannot reproduce the Quasi-Biennial-Oscillation (QBO), thus nudged to QBO reconstruction from Brönnimann et al. (2007). The MA-ECHAM5 (MPIOM) component calculates the dynamical processes in every 15 (144) minutes and atmosphere-ocean coupling takes place in every 24 hours (Anet et al. 2013a, b; Muthers et al. 2014). Muthers et al. 2014 employed SOCOL-MPIOM to carry out four transient simulations (namely L1, L2, M1, and M2) over the period AD 1600-1999 with all major forcings (i.e. greenhouse gases, volcanic eruptions, aerosols, and solar spectral irradiance), and interactive ozone chemistry. The SOCOL-MPIOM was forced with six bands of Solar Spectral Irradiance (SSI) reconstruction of Shapiro et al. (2011) over the Ultraviolet (UV), visible, and near infrared

ranges. The L1 (M1) and L2 (M2) simulations were forced with large (small) mean solar amplitude of 6 (3) W/m2 with different ocean initial conditions for both runs. For more details of the model the reader is referred to Muthers et al. 2014. The model is well capable of simulating the top-down (stratospheric-tropospheric coupling) and bottom-up (coupled ocean-atmosphere response) mechanisms as proposed by Meehl et al. (2009)."

Mean difference (Section 3.1) Do results of figure 5 compare with Fig1 of Ineson et al., 2011? Difficult to say for SLP. For temperature, I see some similarities but some differences as well. I could also consider presenting lagged anomalies (see my following comment).

Answer: The SLP and temperatures differences in Figure 5 are similar to Figure 1 of Ineson et al. (2011) with some variations in the location of the maximum differences. Differences are similar over Europe but quite different over the North Atlantic and Greenland. For example the positive SLP difference over Scandinavia in Figure 5 does not extend as far east (over Greenland) as in Ineson et al. (2011)

Weather type classification (Section 2.2) This section assumes a reader familiar with the different weather types and their within type differences. I am afraid this won't be the case for most of the CP readers. For example, what does the "well discriminated types (P4 L 31)" mean? Or, "days with probability higher than 75%". I think a concise description of the main characteristics of the weather types is needed.

Answer: Since the submission of this manuscript the paper describing the weather types and reconstruction method has been published online (Schwander et al., 2017). We do not want to describe all the method again in this paper but we will improve the description of the weather types to make it more understandable. The probability refers to the method of reconstruction, it's just an indication on the quality of the reconstruction since there is no comparison possible with another weather types time series over such a long period. The reader should look at Schwander et al. (2017) for more information.

Lagged responses The authors in P12 3rd paragraph, briefly discuss the lagged response of westerly types and try to compare with Gray et al. and Thieblemont et al., results. Same in P8 last paragraph. Inferring time lags is very interesting subject and I would recommend a proper presentation, dedicating, perhaps, even a new Section. This could be a valuable contribution to the number of recent papers discussing time lags as they can highlight the importance of atmosphere-ocean coupling.

Answer: Since the strongest signal in weather types occurrences is found without any lag we decided not to focus on lags. However, we will add histograms with a 1, 2 and 3-year lags as a supplementary. Also we will extend the discussion and comparison on the lags.

Some additional considerations, P1. L27: stratospheric ozone + "and heating".

Answer: Will be added.

P2. L10: "phase lag is expected": Perhaps this is not true by the sole action of "top-down" mechanisms. An atmosphere-ocean coupling is required for lags longer than one year at least.

Answer: We agree, we will reformulate the sentence, we speak here more about a lag of a few months.

P2 L20: found a response

Answer: Thank you, will be corrected.

P3. L3: do you mean Gray et al., 2010?

Answer: Yes, it makes more sense to cite a more recent paper.

P3 L4: This is hardly true. Gray et al, show surface signals.

Answer: Will be corrected.

P3. I think the second paragraph should also be extended by discussing results of more

recent model intercomparison such as CCMVal or SolarMIP. See (Austin et al., 2008; Hood et al., 2015; Misios et al., 2015; Mitchell et al., 2015) and references therein.

Answer: Thank you for the suggestion, we will complete the discussion with more recent references.

P3 L25: "It allows us ... weather statistics". Is this true? What is the main difference to Huth et al., 2008b?

Answer: The difference to Huth et al. (2008b) is that we have almost 250 years of daily weather types ($\sim$50 years for Huth). We will modify the sentence to mention that we have a longer time series of data.

P4 L4: Description here is rather confusing. You should clarify that you analyze a merged dataset and not ERA-40 and ERA-int separately. Please elaborate how stitching was performed.

Answer: Yes we realize that the description is not clear enough. We will rewrite and add more information on the reanalysis data.

P4 L17: Is it one of the revised products of sunspot numbers?

Answer: Yes, we will add this information.

P4 L29: "from 1958 to 1998". Why not till 2009?

Answer: Because some of the instrumental data used for the reconstruction stop in 1998. The reference was taken over a period where all data were available (see Schwander et al., 2017).

P6 L25: $CO_2$, $CH_4$, $N_2O$ (subscripts)

Answer: Will be added.

P7 L6: A quantitative difference of the forcing, long term and 11-yr cycle, should be given here.

Answer: Thank you for the suggestion, we will add this information.

P7 L10: ...66th thresholds of sunspot numbers?

Answer: Yes, the same months were selected based on the sunspot number.

P7 L11: Still not clear how percentiles are calculated. Have you subtracted the 11-yr solar cycle before?

Answer: When we speak about the 11-yr solar cycle we always speak about the monthly sunspot number on which the percentiles where computed. The was used also for the Shapiro reconstruction and is visible in the reconstruction although it is sometime masked by the low frequency variability. We will probably focus only on the period 1958-1999 in the model simulations as a comparison with the reanalysis data. Also the low frequency variability of the solar variability during the period 1958-1999 is stable and we can focus only on the 11-yr solar cycle.

P11-13: It is very difficult to follow the discussion of the results. Please point to the associated figures.

Answer: The discussion will be rewritten to be more understandable and completed with the suggestions from both reviewers.

P13, L11: "only partially". This is a wishful thinking!

Answer: We can remove this, it is true that we do not see the same signal in model simulations.

Figure 5: Difficult to separate SLP from geopotential signals. Please consider splitting this panel in two.

Answer: We will redo this Figure by splitting it or by changing the colors.

Austin, J. et al., 2008. Coupled chemistry climate model simulations of the solar cycle in ozone and temperature. J Geophys Res-Atmos, 113(D11): D11306. Hood, L. et

al., 2015. Solar Signals in CMIP-5 Simulations: The Ozone Response. Q. J. Roy. Meteorol. Soc., 141: 2670–2689. Misios, S. et al., 2015. Solar Signals in CMIP5 Simulations: Effects of Atmospherre Ocean Coupling. Q. J. Roy. Meteorol. Soc. Mitchell, D.M. et al., 2015. Solar Signals in CMIP-5 Simulations: The Stratospheric Pathway. Q. J. Roy. Meteorol. Soc., 141: 2390-2403.

References

Anet. J. G., Muthers, S., Rozanov, E. V., Raible, C. C., Peter, T., Stenke, A., Shapiro, A. I., Beer, J., Steinhilber, F., Brönnimann, S., Arfeuille, F. X., Brugnara, Y., Schmutz, W.: Forcing of stratospheric chemistry and dynamics during the Dalton Minimum. Atmos Chem Phys 13:10951–10967. doi:10.5194/acp-13-10951-2013, 2013a.

Anet, J. G., Rozanov, E. V., Muthers, S., Peter, T., Brönnimann, S., Arfeuille, F. X., Beer, J., Shapiro, A. I., Raible, C. C., Steinhilber, F., Schmutz, W. K.: Impact of a potential 21st century "grand solar minimum" on surface temperatures and stratospheric ozone. Geophys Res Lett 40(16). doi:10.1002/grl.50806, 2013b.

Brönnimann, S., Annis, J. L., Vogler, C., Jones, P. D.: Reconstructing the quasi-biennial oscillation back to the early 1900s. Geophys Res Lett 34(L22805). doi:10.1029/2007GL031354, 2007.

Egorova, T., Rozanov, E., Zubov, V. and Karol, I. L.: Model for investigating ozone trends (MEZON), Izv. Atmos. Ocean. Phys., 39, 277–292, 2003.

Meehl, G. A., Arblaster, J. M., Matthes, K., Sassi, F., Loon, H.V.: Amplifying the Pacific climate system response to a small 11-year solar cycle forcing. Science 325(5944). doi:10.1126/science.1172872, 2009.

Muthers, S., Anet, J. G., Stenke, A., Raible, C. C., Rozanov, E., Brönnimann, S., Peter, T., Arfeuille, F. X., Shapiro, A. I., Beer, J., Steinhilber, F., Brugnara, Y. and Schmutz, W.: The coupled atmosphere-chemistry-ocean model SOCOL-MPIOM, 5 Geosci. Model Dev., 7(5), 2157–2179, doi:10.5194/gmd-7-2157-2014, 2014

Roeckner, E., Bäuml, G., Bonaventura, L., Brokopf, R., Esch, M., Giorgetta, M., Hagemann, S., Kirchner, I., Kornblueh, L., 20 Rhodin, A., Schlese, U., Schulzweida, U. and Tompkins, A.: The atmospheric general circulation model ECHAM5: Part 1: Model description, MPI Rep., (349), 1–140, doi:10.1029/2010JD014036, 2003.

Schwander, M., Brönnimann, S., Delaygue. G., Rohrer. M., Auchmann, R., Brugnara, Y. Reconstruction of Central European daily weather types back to 1763. International Journal of Climatology. doi: 10.1002/joc.4974. 2017.

Weusthoff, T.: Weather Type Classification at MeteoSwiss - Introduction of new automatic classification schemes, Arbeitsberichte der MeteoSchweiz, (235), 46, 2011.

---

## Author Response (AR1)

**Author's response**

**Review 1**

**The manuscript presents a detailed investigation of 11-yr solar cycle influence on weather types. Thetopicisofinterestforthepaleoclimateandsolar-terrestrialcommunities, although a detailed knowledge of the weather types and their variability may be too farfetched for most of the CP readers. Some description of weather type characteristics is needed. Besides a new, unique reconstruction of weather types, the authors analyze a set of 4 simulations with a climate model forced by Total Solar Irradiance, only. This is a somewhat a strange model configuration and weakens the merit of the simulations for detecting mechanisms, as the "top-down", which believed to be the key mechanism influencing the weather types (see Introduction) is neglected. Most parts of the manuscript are well written and reading is straightforward and easy, except the discussion part where it is difficult to pair results of this study to the text. There are some points other that need further clarification and improvement before publication of the paper.**

We thank the reviewer for the constructive comments. We agree on most comments and included the suggestions in the revised manuscript. The description of the model in our first manuscript was too vague and was improved in the new version since the model is forced by SSI. We decided to leave out the low frequency solar variability of the model since it is not in the scope of the papers and only brings confusion. We focus only on the 11-year solar cycle from 1958 to 1999 to compare with reanalysis data. The model includes bottom-up and top-down mechanisms.

**Methodology**

**Weather types are analyzed by the means of composites. The 11-yr solar cycle is sliced in three groups/day of low, moderate and high activity. Are these groups of near equal size? I am concerned about the role of internal variability in the composites and how affects results. How confident are the authors that compositing results in true solar signals? Splitting the record to 50 yr chunks offers too little to this regard because it provides little evidence of consistency over time. This is recognized by the authors: "P8 L13 Although it can be difficult ...". I would suggest to split to two sub-periods at best. For the same reason, Figure 7 can be supplied as a supplementary.**

The groups are not exactly of equal sizes since more volcanic eruptions occurred under low solar activity. We added a table with the size of the groups. The confidence is quite high since the signal in some types in significant over 250 years, and also because it is consistent with previous studies. The signal found in the occurrence of weather types is consistent with the within-type differences. The significance was added on the within-type composites difference plots.

We kept the 1958-2009 sub-periods as a comparison with Huth et al. (2008). We reduced the number of sub-periods to have only two (1763-1886 and 1887-2009).

**Significance in solar minimum**

**I always consider the solar minimum as the least perturbed state of climate not necessarily the reverse of the solar maximum. It is puzzling to me that the most noticeable changes in WSW and W types are detected in solar minimum, when the forcing is weakest. Could the authors elaborate on the reasons/mechanisms that can explain strongest signals in solar minimum and not maximum?**

We cannot provide any reasons why the signal could be stronger under low solar activity compared to high activity. We think that it comes from the fact that the long-term mean is also perturbed. If the low activity phase is not perturbed (one third of the months), then two thirds are perturbed (moderate and high). The long-term mean is therefore also perturbed and the differences under low solar activity seem larger. We could take the low solar activity phase as a (unperturbed) reference and then we would observe a strong increase in the occurrence of W and WSW types under moderate and high solar activity.

**Model simulations**

**Perhaps I am missing something here, but my understanding is that the SOCOL simulations are forced only by TSI and in particular by the strong Sapiro et al. TSI reconstruction. There is nothing wrong by choosing a strong TSI reduction to facilitate the signal-to-noise detection. My objection here is on the specification of TSI and not SSI variability. Is there any particular reason to assume that solar signals in weather types are attributed to the "bottom-up" mechanisms? Most of the discussion in introduction emphasizes the importance of "top-down" mechanisms in transferring signals on the surface, a mechanism which apparently is missing in model runs without SSI forcing. In such a case, the low resemblance between reanalysis and modelled signals is not surprising to my understanding. Moreover, some similarities discussed in P.10 L30 is a matter of coincidence to me. So, it is difficult to understand the overall point of Section 3.4 given that the SOCOL runs are missing key mechanisms. The weakness of the simulations should be discussed in the text.**

We realize that the model description in the Manuscript is lacking some information. The model was not forced by SSI and can include top-down and bottom-up mechanisms. We have rewritten the description of the model (Section 2.6). We now only focus on the 11-year cycle and have removed the part on the low frequency of the solar variability.

**Mean difference (Section 3.1)**

**Do results of figure 5 compare with Fig1 of Ineson et al., 2011? Difficult to say for SLP. For temperature, I see some similarities but some differences as well. I could also consider presenting lagged anomalies (see my following comment).**

The SLP and temperatures differences in Figure 5 (now Figure 4) are similar to Figure 1 of Ineson et al. (2011) with some variations in the location of the maximum differences. Differences are similar over Europe but quite different over the North

Atlantic and Greenland. For example the positive SLP difference over Scandinavia in Figure 5 (now Figure 4) does not extend as far east (over Greenland) as in Ineson et al. (2011)

**Weather type classification (Section 2.2)**

**This section assumes a reader familiar with the different weather types and their within type differences. I am afraid this won't be the case for most of the CP readers. For example, what does the "well discriminated types (P4 L 31)" mean? Or, "days with probability higher than 75%". I think a concise description of the main characteristics of the weather types is needed.**

Since the submission of the first manuscript the paper describing the weather types and reconstruction method has been published online (Schwander et al., 2017). We do not want to describe all the method again in this paper but tried to improve a little the description of the weather types to make it more understandable. The probability refers to the method of reconstruction, it's just an indication on the quality of the reconstruction since there is no comparison possible with another weather types time series over such a long period. The reader should look at Schwander et al. (2017) for more information.

**Lagged responses**

**The authors in P12 3rd paragraph, briefly discuss the lagged response of westerly types and try to compare with Gray et al. and Thieblemont et al., results. Same in P8 last paragraph. Inferring time lags is very interesting subject and I would recommend a proper presentation, dedicating, perhaps, even a new Section. This could be a valuable contribution to the number of recent papers discussing time lags as they can highlight the importance of atmosphere-ocean coupling.**

Since the strongest signal in weather types occurrences is found without any lag we decided not to focus on lags. However, we added a new figure with lags (Figure 6) and extended slightly the discussion.

**Some additional considerations,**

**P1. L27: stratospheric ozone + "and heating".**

We have made the correction.

**P2. L10: "phase lag is expected": Perhaps this is not true by the sole action of "topdown" mechanisms. An atmosphere-ocean coupling is required for lags longer than one year at least.**

We agree, we have mentioned that it is only a short lag in this case.

**P2 L20: found a response**

Thank you, we have corrected this.

**P3. L3: do you mean Gray et al., 2010?**

Yes

**P3 L4: This is hardly true. Gray et al, show surface signals.**

5   We have reformulated the sentence.

**P3. I think the second paragraph should also be extended by discussing results of more recent model intercomparison such as CCMVal or SolarMIP. See (Austin et al., 2008; Hood et al., 2015; Misios et al., 2015; Mitchell et al., 2015) and references therein.**

10   Thank you for the suggestion, the discussion have been completed with some more references.

**P3 L25: "It allows us ... weather statistics". Is this true? What is the main difference to Huth et al., 2008b?**

The difference to Huth et al. (2008b) is that we have almost 250 years of daily weather types (~50 years for Huth). We have modified the sentence to mention that we have a longer time series of data.

**P4 L4: Description here is rather confusing. You should clarify that you analyze a merged dataset and not ERA-40 and ERA-int separately. Please elaborate how stitching was performed.**

We have made the description of ERA-40 and ERA-Interim clearer.

20   **P4 L17: Is it one of the revised products of sunspot numbers?**

Yes, we have added this information.

**P4 L29: "from 1958 to 1998". Why not till 2009?**

Because some of the instrumental data used for the reconstruction stop in 1998. The reference was taken over a period where

25   all data were available (see Schwander et al., 2017).

**P6 L25: CO2, CH4, N2O (subscripts)**

We have added the subscripts.

30   **P7 L6: A quantitative difference of the forcing, long term and 11-yr cycle, should be given here.**

We have rewritten the section on the model simulations, and we now only focus on the 11-year cycle.

**P7 L10: ...66th thresholds of sunspot numbers?**

Yes, we have completed the sentence.

**P7 L11: Still not clear how percentiles are calculated. Have you subtracted the 11-yr solar cycle before?**

When we speak about the 11-yr solar cycle we always speak about the monthly sunspot number on which the percentiles where computed. The was used also for the Shapiro reconstruction and is visible in the reconstruction although it is

5   sometime masked by the low frequency variability. We have decided to focus only on the period 1958-1999 in the model simulations as a comparison with the reanalysis data. The low frequency variability of the solar variability during the period 1958-1999 is stable and we can focus only on the 11-yr solar cycle.

**P11-13: It is very difficult to follow the discussion of the results. Please point to the associated figures.**

10  We have rewritten some parts of the discussion and pointed to the corresponding figures.

**P13, L11: "only partially". This is a wishful thinking!**

We have removed this, it is true that we do not see the same signal in model simulations.

15  **Figure 5: Difficult to separate SLP from geopotential signals. Please consider splitting this panel in two.**

We did not split the figures but we have adapted it to make it more understable.

**Austin, J. et al., 2008. Coupled chemistry climate model simulations of the solar cycle in ozone and temperature. J Geophys Res-Atmos, 113(D11): D11306. Hood, L. et al., 2015. Solar Signals in CMIP-5 Simulations: The Ozone**

20  **Response. Q. J. Roy. Meteorol. Soc., 141: 2670–2689. Misios, S. et al., 2015. Solar Signals in CMIP5 Simulations: Effects of Atmospherre Ocean Coupling. Q. J. Roy. Meteorol. Soc. Mitchell, D.M. et al., 2015. Solar Signals in CMIP-5 Simulations: The Stratospheric Pathway. Q. J. Roy. Meteorol. Soc., 141: 2390-2403.**

**Section 2.1 should be clearly written, and the indices they used for the 11-year and longer term variability presented in a very clear way. For example, there is no call to Figure 3 in this section. They refer to Figure 4 but with no explanation as to what it contains, and the reader is left puzzled, since the Shapiro reconstruction is shown there without it being mentioned in the text. Moreover, I could not understand why they mention in the text that the fact that the sunspot cycle does not become negative is a limitation (this is also mentioned again later in the paper).**

We agree on the comment; this section has been improved and the corresponding figures are now mentioned and better presented.

**Section 2.3**

**It is not clear what are the time scales they discuss. Do they refer to the 11-year of the secular cycles? This should be very clearly mentioned here as well. The mechanism they refer to is the top-down mechanism, in which the stratospheric response and the signal transfer from there to the troposphere is the main pathway. This leads us to**

We focus only on the 11-year solar cycle.

**Section 2.6, where they describe the model simulations. Again in line 21 they refer to low and high solar activity, with no clear indication as to what they mean. Moreover, and for the model simulations: Was TSI the only forcing? Or did they use also the appropriate SSI forcing? Was the model run in its full version with the interactive ozone response in the stratosphere? How is it achieved if one uses TSI variations only? Was the solar effect on ozone included in any way? If SSI variability with the solar cycle and the stratospheric response is not included, then one can have only the bottom up mechanism, and the comparison to e.g. Ineson et al. is not straight forward. In addition, what is the meaning of "It has the advantage to be a predominant forcing in the model.."? It is also not clear how the 11-year solar cycle is handled here. The Shapiro index and its use to define "large solar activity", "moderate amplitude" should be more clearly written.**

We have realized that the description of the model simulations in the manuscript was not clear enough. SSI is included in the model. We have rewritten the paragraph presenting the model simulations.

"It has the advantage to be a predominant forcing in the model.." means that since the Shapiro reconstruction has a higher amplitude (~6 w/m$^2$) than any other reconstruction, it consists of a strong forcing in the model. The upper boundary of the uncertainty of the Shapiro reconstruction was used as moderate amplitude (~3 w/m$^2$) in the model. Also the Shapiro reconstruction includes the 11-yr solar cycle (based on the sunspot number) although it is often masked by the low frequency amplitude.

The analysis of the low frequency solar activity in the model did not bring any relevant conclusions to the papers. So we have decided to focus only on the period 1958-1999 in the model simulations as a comparison with the reanalysis data. Also the low frequency variability of the solar variability during the period 1958-1999 is stable and we can focus only on the 11-yr solar cycle.

**Page7 line9-10, on the volcanic activity and the years that were removed. Why do you state there to "note that many of the important eruptions occur during a solar minimum". Is there any possible connection? How does the removal affect your statistics if it was mainly done for solar minimum years? And more importantly, what type of solar minimum? Sunspot, or secular?**

We are not aware about any connection between volcanic eruptions and the solar cycle. The fact that more volcanic eruptions occurred under low solar activity phases has mostly an impact on the size of (number of months) of the low solar activity class. We have added a table with the size of each group (table 2).

**Page 7, lines 15 -18. How exactly was the anthropogenic forcing removed? What were the predictors? Was there only one predictor? Which one?**

The predictor consists in the radiative forcing applied in the model calculated from major greenhouse gases (CO2, CH4, N2O and CFCs). They were taken from the PMIP3 database (Etheridge et al., 1996, 1998; Ferretti et al., 2005; MacFarling-Meure et al., 2006).

**Section 3.3 Significance in the differences should be given. The same holds for every place where differences are discussed.**

We have added significance on the differences plots. Also we have corrected it in Figure 5 (now Figure 4) since we have found a small error in the significance plotted.

**4 Discussion Page 11, lines 18-19. It is accepted that the 11-year cycle effects project onto tropospheric circulation patterns like the Arctic Oscillation (AO) and the North Atlantic Oscillation (NAO) rather than are directly correlated to NAO or AO**

We have reformulated the sentence to make it clear that we are not speaking about a direct correlation.

**5. Conclusions page 14, lines 4-6. The present simulation and the forcings used (if indeed SSI variability and ozone related variability have not been used) do not allow the investigation of the top-down mechanism, which is in the heart of the weather type response..**

SSI and the related ozone variability are included in the model.

In the introduction, we have added some reference as suggested by both reviewers.

We have improved the description of the data and methods since some information was missing on the ERA-40 and ERA-Interim and on the model. We added some sentences to make the weather types description clearer.

The number of sub-periods for the histograms (Fig. 5) was reduced and we have added a figure focuses on the lags (Fig. 6) as suggested by one of the reviewers.

Significance have been added on difference plots (Fig. 4, 7 and 8).

A table was added with the size of each group (number of months) for the different periods analyzed (Table 2.).

The discussion was adapted and made clearer.

Most of the figures were computed again since there was a small error in the data (some volcanic years have not been removed). However, it has no impact on the discussion and conclusions of the paper.

Since we have decided not to focus on the low frequency, we have remove Fig. 4, 11 and 12 (from the first manuscript).

[revised manuscript text omitted]

---

## Author Response (AR2)

**Author's response**

**Report #1 / Anonymous Referee #2**

**Review of the revised paper**

5 **"Influence of solar variability on the occurrence of Central European weather types from 1763 to 2009" by Mikhaël Schwander et al.,**

**The paper can be accepted at its present form. However, it needs to be checked for spelling/grammar.**

We thank the reviewer for the comments. We have checked for spelling and grammar mistakes.

10 **I have only some minor/technical comments, e.g.**

**-Page 7, line 10:**

**"The SOCOL-MPIOM cannot reproduce" please change to "does not reproduce…"**

As suggested by the other referee we have removed the part focusing on the model simulations since it does not bring any

15 relevant conclusion to the paper.

**-Figure captions of Figs 3, 4, 7, 8 and 9 and also in all appropriate places in the text:**

**Please indicate units everywhere (also for blocking frequencies)**

20 We have added the units in the figures captions and in the text when needed.

**Report #2 / Anonymous Referee #1**

**I appreciate the effort taken by the authors to revise the manuscript, which reads much better now. Yet, there are some points that need further improvement to make manuscript more concise and deliver a clearer message. My recomendation for major revision is mostly related to the part of model results.**

5  We thank the reviewer for the constructive comments. We have made corrections following the suggestions.

**Merit of simulations.**

**Reading the manuscript in the present form, I would suggest to take out the part of model simulations. Section 3.4 where some model results are described is very short (I count 7 lines!) and do not corroborate the observed evidence.**

10  **I would partly agree/disagree on the conclusion that model resolution is too coarse to simulate large scale changes associated with the NAO, but even if it is the case, what is the point of using such a model?**

We agree that the model simulations do not bring any relevant conclusions to the paper. As suggested we have taken this part out.

15  **Presentation of solar signals**

**In some parts, you note that MAX-MIN are subtracted but in other you present MIN-MAX. Please make it consistent throughout the text.**

We have not found any inconsistencies. In all cases is MIN-MAX presented. In the text it is written as "low minus high" or "subtracting the high activity from the low activity".

**Positive or Negative NAO?**

**In the discussion, you note that results resemble a negative NAO in solar MIN, but immediately after you mention that NAO get positive. I wouldn't say from SLP anomalies (figure 4.a) that signal matches any phase of NAO.**

Thank you for the relevant comment, it is true that the text is not clear and the signal does not really match any phase of

25  NAO. We have made the necessary corrections in the text.

**Relation between figure 5 and figure 4**

**In p24 l. 9 you attribute signals in Figure 4 to changes in figure 5. Figure 4 refers however to the 1958-2009 period during which there is no significant change in W type (figure 5c). There is a drop in WSW but not significant.**

30  **Likewise, a stronger increase in E type (still insignificant) is seen for the moderate compared to low years over 1958-2009. I don't think that your argument here is backed by results.**

We agree that the comparison between figure 5 and 4 in the text is not totally correct. We have adapted this part. However, we still think that the difference between low and high solar activity in the frequency of occurrence (Fig. 5, 1958-2009) of

the NE, WSW, E and N types (even if not significant) are in agreement with the mean difference (Fig. 4, reduced zonal flow, lower temperature).

**Other (counting is based on the version of the manuscript with highlighted changes stitched with the responses to reviewers.)**

**p11 (first page), l.25: … atmospheric circulation over Europe?**

Yes, we have added this clarification.

**p12 , l.2 : please add reference list in chorological order**

According to the CP " Manuscript preparation guidelines for authors": "In terms of in-text citations, the order can be based on relevance, as well as chronological or alphabetical listing, depending on the author's preference.".

**p12, l. 5: is Sitnov relevant? Soukharev and Hood, 2006 is relevant for the solar cycle signals in ozone.**

We have added Soukharev and Hood (2006). Sitnov (2009) also focuses on the ozone in the stratosphere and the downward propagation.

**P12, l.25: please rewrite. Enhanced references to the strength not the phase.**

We have rewritten this sentence.

**P12. L26: … influence projects …**

Corrected.

**P12. Second paragraph: too many "found". Please rewrite**

We have rewritten parts of the paragraph.

**P12 l.31: Why NAO effects could be visible only on short time scales?**

We have corrected this sentence. We did not want to say that the effects are "only visible on short time scales", but "also visible on short time scales". We have changed the paragraph to say that the effects are visible on weather patterns.

**P13 l. 1: I think the results of Sirocko et al., have been questioned on statistical grounds**

We mention two paragraphs later that the method of Sirocko et al. shows some weaknesses as explained in van Oldenborgh et al. (2013).

**P13 l. 14: define CMIP5**

We have added the full name.

**P13 l. 15: My impression is that Mitchell et al., DID NOT demonstrate any significant lagged NAO changes. Misios et al. has nothing to do with the polar night jet.**

Mitchel et al. (2015) say that they have found a lag in the North Atlantic but with a weaker response than in the observations. "The lagged North Atlantic surface response is reproduced in both high- and low-top models, but is more prevalent in the former. In both cases, the magnitude of the response is generally lower than in observations"

Thank you for pointing this out. It is a mistake; this sentence does not refer to Misios et al. (2016) but to Hood et al. (2015).

**Section 2.1 : too much information**

We have shortened this Section.

**P15 l. 10: … used as a reference …**

Corrected.

**P15 l. 11: not well discriminated types: what do you mean?**

We have added "pairs of types having a similar mean slp pattern". The reader should see Schwander et al. (2017) for more information.

**P15 l. 13: 7 -> seven**

Corrected.

**P16 l. 22-24: it should go to Data and Methods**

We have moved this sentence to Section 2.

**P17 l 5: Earlier you mention 1x1 grid size. Here you interpolate only ERA-40? What about ERA-int?**

For the computation of blockings, both ERA-40 and ERA-Interim are used with a 2x2 grid size. We have added this information in the text.

**P17 l. 28: The SOCOL -> SOCOL (see also my comment about taking out the model simulations)**

We have removed the part on the model simulations as suggested.

**P19 l. 17: For the last 50 years (Figure 5c)….**

Corrected.

**P21 l.1: … analysis is complemented? ….**

Corrected.

**P24 l. 1: and is visible -> and it still persists?**

Corrected.

**P24 l. 17: are similar at zero lag**

Corrected.

**P26 l. 15: Following these….-> Based on this observational evidence,**

Corrected.

**List of relevant changes made in the manuscript**

As suggested by referee #1 we have removed all parts focusing on the model simulations (incl. Fig. 9). We agree that it does not add any important information to the paper or any relevant conclusion.

Some corrections have been made in the discussion part relative to the signal resembling (or not) to any NAO phase.

Spelling/grammar mistakes have been corrected.

[revised manuscript text omitted]

---

## Author Response (AR3)

**Author's response**

**Review by Editor**

**I have just two more points i wish you could address before the paper can be accepted.**

Thank you for these comments.

**The first one relates to the point mentioned by reviewer 2 about the Sirocko et al paper and its clear flaws that have been clearly detailled in van Oldenborgh et al. This needs to be pointed out in more detail**
**The current formulations and not yet adequate.**

We have completed the part presenting van Oldenborgh et al. (2013) in the introduction with a short description of the three problems that they found in Sirocko et al. (2012).

**The second point is that the analysis ends in 2009. Why did the last years not included?**

The CAP7 weather type classification ends in 2009 because some of the instrumental data time series do not extend further in time. We could have used the CAP9 weather types from MeteoSwiss further on but this could add some inhomogeneity in the study. Also for the 1763-2009 analysis the addition of a few years of data would probably not significantly change the main findings.

We have added a sentence in Section 2.2 to explain why CAP7 stops in 2009.

**5 List of relevant changes made in the manuscript**

See above.

[revised manuscript text omitted]